**communications** engineering

# Rapid in-air ultrasound holography measurement and camera-in-the-loop generation using thermography
Zak Morgan ⊠, Youngjun Cho & Sriram Subramanian

Ultrasound holography, pivotal in applications like mid-air haptics, volumetric displays and 3D printing, faces challenges in the crafting and timely measurement of precise acoustic holograms. Current methods, bench-marked via simulations due to slow measurement times, often neglect real-world complexities such as non-linearity and hardware tolerances, leading to discrepancies between predicted and observed results. Here we introduce a real-time 2D thermographic measurement technique orders of magnitude faster than microphone scans, although with reduced accuracy and no phase information, with a maximum peak pressure of 4.25 kPa validated and a demonstrated average accuracy of 2.5% in peak measurement. Higher pressures of approximately 12 kpa were captured, but validation was limited by the microphone. This method is grounded in thermo-viscous acoustic models for thin-ducts and micro-perforated plates. Finally, we integrate this with holography algorithms to propose a camera-in-the-loop algorithm that employs real-time measurement, enabling targeted data acquisition and on-line training of acoustic holography algorithms. This method achieved a 1.7% error in pressure with a single point compared to 7.8% for a conventional algorithm, and a 3.6% error and 4.2% standard deviation for 16 points compared to 9.7% and 6.9%. We further envisage this method as being capable of measuring acoustic streaming.

With the rise of ultrasonic applications like mid-air haptics[1], volumetric displays[2] and contact free fabrication[3], crafting precise in-air acoustic holograms (a specific spatial distribution of sound pressure) is rapidly becoming a field of interest. That is, given a desired hologram in a target plane (or volume), how can we control an acoustic device in such a way that it produces such an output. This process of crafting an input at the hologram plane (the acoustic device plane), which propagates through space and recreates a target at the image plane is known as computer-generated holography[4]. This is both an ill-conditioned[5] and ill-posed inverse problem[6] as solutions are not necessarily unique unlike the phase-retrieval problem[7] and small changes in input can offer large changes in output. These problems motivate the field of non-convex optimisation in optics[8], however in the acoustic domain there are further problems to contend with. The first is the lack of a camera analogue, making measuring holograms across a 2D area a much harder task. The second is the lack of an equivalent of a spatial light modulator or lens, for controlling propagation direction, amplitude and phase through a medium, with phased arrays commonly used as a substitute in their place. Finally there is also the stronger influence of non-linear effects on propagation[9].

In this paper we take aim at the first problem, and by realising it, provide a route for transition of some techniques from the optical domain to acoustics, such as on-line computer-generated holography[10]. Whilst it is clear it is indeed a problem, it is not immediately obvious what the consequences of it are. In order to avoid this problem, previous works often rely on simulations to measure the performance of state-of-the-art acoustic holography algorithms[11–13]. A consequence of this is that they often neglect non-linear effects[9], ambient and device conditions such as temperature[14,15] and humidity, as well as specific device properties such as the impulse and frequency response of transducers. This can result in consequences such as acoustic traps not exhibiting the predicted strength, leading to levitated particles dropping[15]. Recent advances in computer-generated holography algorithms have started to account for some of these properties like impulse response for moving levitated objects[16], although measuring the impact of these on the actual holograms is not done, limiting their evaluation to simply the end result of levitation, limiting understanding of their effectiveness.

For perspective, in optics and hydro-acoustics, it is a more common practice to blend simulated (analytical) data with real measurements to achieve more reliable results such as in ref. 17 or when calibrating spatial light modulators[18]. Even a small model miss-match can cause large errors in conventional algorithms based on techniques like the family of GS (Gerchberg-Saxton) algorithms and those based on Wirtinger flow

Department of Computer Science, University College London, London, UK. ⊠e-mail: zak.morgan.17@ucl.ac.uk

derivatives[10]. This challenge of replacing or augmenting these simulations with empirical measurements is what this paper helps to alleviate, by presenting an analogue to a camera in acoustics.

The main current method used in acoustics is the scanning microphone, effectively measuring 1 'pixel' or 'voxel' at a time before being moved in space. The term pixel or voxel is used here since measurements of sound-fields are often displayed as 2D images, as slices of some 3D sound field, where the colour of each pixel represents amplitude or phase, with the x and y position representing spatial location, where all measurements take place on some uniform grid. Whilst a microphone array can be used instead of a single microphone to increase the resolution, the cost of such a system rapidly becomes high, as does the electronics required to record a signal from each of them. This scanning technique is also used in optics for high-resolution photography, commonly called pixel-shift super-resolution[19]. Due to the limited resolution of each sample, measuring acoustic holograms is often very slow, taking approximately one second per sample[20] and thus tens of minutes to hours per scan. This also hinders performance evaluation and accurate measurement in the case of dynamic or unstable sound fields, as well as preventing quick algorithmic iteration. In addition to time, there is also a problem in measuring large pressures. The two main commercial microphones used for in-air ultrasound measurement are the B&K 4138 and GRAS 40DP, which cannot accurately measure above a 168 and 178 dB sound pressure level, respectively, of which is quite easily surpassed when focusing using a large amount of transducers.

Other alternatives to microphones are optical methods such as schlieren[21,22], laser Doppler vibrometer[23], diffraction[24], heterodyne interferometer[25] and fabry-perot sensors[20] to name a few. All of these methods require optical access, long working distances, and specialized and often expensive setups. In addition multiple measurements from different angles are usually required and then combined using tomographic reconstruction to produce the final result, although recent advances in optical methods such as phase-shifting digital holography[26] remove the requirement for computer tomography algorithms and thus allow single shot measurement. In addition large amplitudes are difficult to measure as they require phase-unwrapping algorithms, such as those in ref. 27 which are far from perfect. Despite this these methods are however much more sensitive than microphones and the thermographic based measurement presented in this paper, and additionally can offer measurement of frequencies far above microphones, which is crucial in work that utilises these higher frequencies such as for investigating non-linear effects which presents in higher-order harmonics[9].

A final alternative is to transform the acoustic energy into some quantity which is more easily measurable, otherwise known as transduction, and the device a transducer. This is the principle microphones act on, by transforming acoustic energy into an electric current. It is also possible to place an acoustic absorber in a field, which transforms acoustic energy into heat or even light through acoustically induced piezo-luminescence[28,29]. By measuring this energy it is possible therefore to construct an acoustic camera. Focussing on the heat case, the most complicated and expensive part of the set-up is the thermal camera. Whilst it is possible to use cheap thermo-chromatic materials to measure this heat increase[30], calibration is difficult[29]. In the hydro-acoustics domain, measuring the steady state temperature of a thin acoustic absorber has been shown to measure a 3D volume of 3 cm × 3 cm × 6 cm in 33 s with a 10 and 6% error in peak pressure for the two focal points present, although the error was not stated for the area around the peak, and it is clear that it is large[31].

Transitioning this method to air introduces a new set of complexities. The primary problem is due to reduction in heat transfer in air (7 $Wm^{-2}K^{-1}$ [32] compared to water >50 $Wm^{-2}K^{-1}$ [33]), which leads to an accumulation of heat dissipation within the measurement material itself, instead of in the surrounding medium. Further the decrease in frequency, density and sound speed as well as larger focal points lead to streaming velocities an order of magnitude higher[34]. At the reported streaming velocity of 2.9 $ms^{-1}$ for a 2 kPa root mean squared (RMS) focal point[35], the heat transfer coefficient would rise from 5 to 20–45 $Wm^{-2}K^{-1}$ depending on the level of

turbulence[36]. Thus in the centre of a focal point for example, it is possible to have an order of magnitude more cooling than its surroundings. This non-homogeneous cooling behaviour leads to difficulties in the steady state model in air. Finally, creating an absorbent material with a similar acoustic impedance to air, is much more difficult than in water, and requires consideration of geometry in order to create a porous absorber instead of matching the speed of sound.

An attempt to bring this method to the in-air domain has been made[37], but it attempted to measure the heat generation in the air itself, rather than in a measurement material, assuming the material temperature increase was identical to that of air. This meant they did not account for any properties of the material, such as its specific heat capacity or geometry. This resulted in poor accuracy levels (Error of 13% in peak pressure of a focal point) in a 1D measurement and with only one data-point. Additionally the measurement material was parallel to the direction of acoustic propagation, thus limiting absorption and sensitivity.

In order to address these problems, we present in this paper a rapid 2D measurement technique utilising thermography, by measuring the temperature increase of a porous absorber When a mono-tonal acoustic hologram is projected onto it, with the structure shown in Fig. 1. Through modelling the absorber as a micro-perforated panel (MPP)[38], the pore size can be carefully selected to maximise its narrowband absorption properties for the frequency of interest. Figure 1 shows all steps in this process, as well as its calibration, and how it is integrated into a camera-in-the-loop (CITL) holography method. A video is first processed by a non-uniformity correction to eliminate fixed-pattern noise in the image, then lens correction and a high-pass filter. The start of the acoustic radiation is then found, the gradient or steady state temperatures are extracted, and a thermal soak and diffusion model is used, before finally mapping these temperature-based metrics to pressure, based on either the gradient, or using one of the steady state models.

A clear advantage of a thermography based method over these competing methods is the small physical size as the camera is only 106 × 47 × 50 mm, whilst optical methods often require large optical tables, beam splitters, lens and other optical equipment. In addition, the simple material and device requirements make the method widely available, with the only requirements being a mesh material which can be had for several dollars, and a thermal camera which can be had for around one to two hundred dollars. In addition to this the simple and robust equipment set-up means any minor mis-alignment of parts will not lead to a complete measurement failure, as opposed to schlieren for instance, where vibration can cause the light beam to miss the knife edge. Finally, the computation done after the fact is simple and does not require any computer tomography algorithms or phase unwrapping, making analysis simple. The primary disadvantage of our method compared to the competing methods is the inability to measure phase. Though another downfall is the limited sensitivity and noise floor, due to limited thermal camera sensitivity, in addition to the fact that small changes in temperature are instantly lost to the environment without being measureable.

We demonstrate not only an impressive accuracy of our system in peak pressure of 2.2%, but crucially across the entire 2D xy (perpendicular to the source) plane of a focal point with a mean root mean square error (RMSE) of 64 Pa. This is achieved through the application of noise reduction algorithms and heat modelling, leading to a 30–50% increase in accuracy when used and can be seen in the differences in Fig. 2. We also demonstrate 2D metrics such as a sub-mm error in Full Width at Half Maximum (FWHM), crucial for work in haptics and large-object levitation, where acoustic radiation force is the measured quantity[39–41] which is proportional to pressure integrated over an area. In addition, our system's computational cost is shown to be low enough to run measurements in real-time. Finally, we demonstrate its application in a simple Proportional-Integral-Derivative based CITL system showcasing an accuracy of 3.6% and standard deviation of 4.2% between focal points in a multi-point scenario compared to 9.7% and 6.9% in the under-lying algorithm without extensive device calibration. We further hypothesise that this technique can also be used for measuring

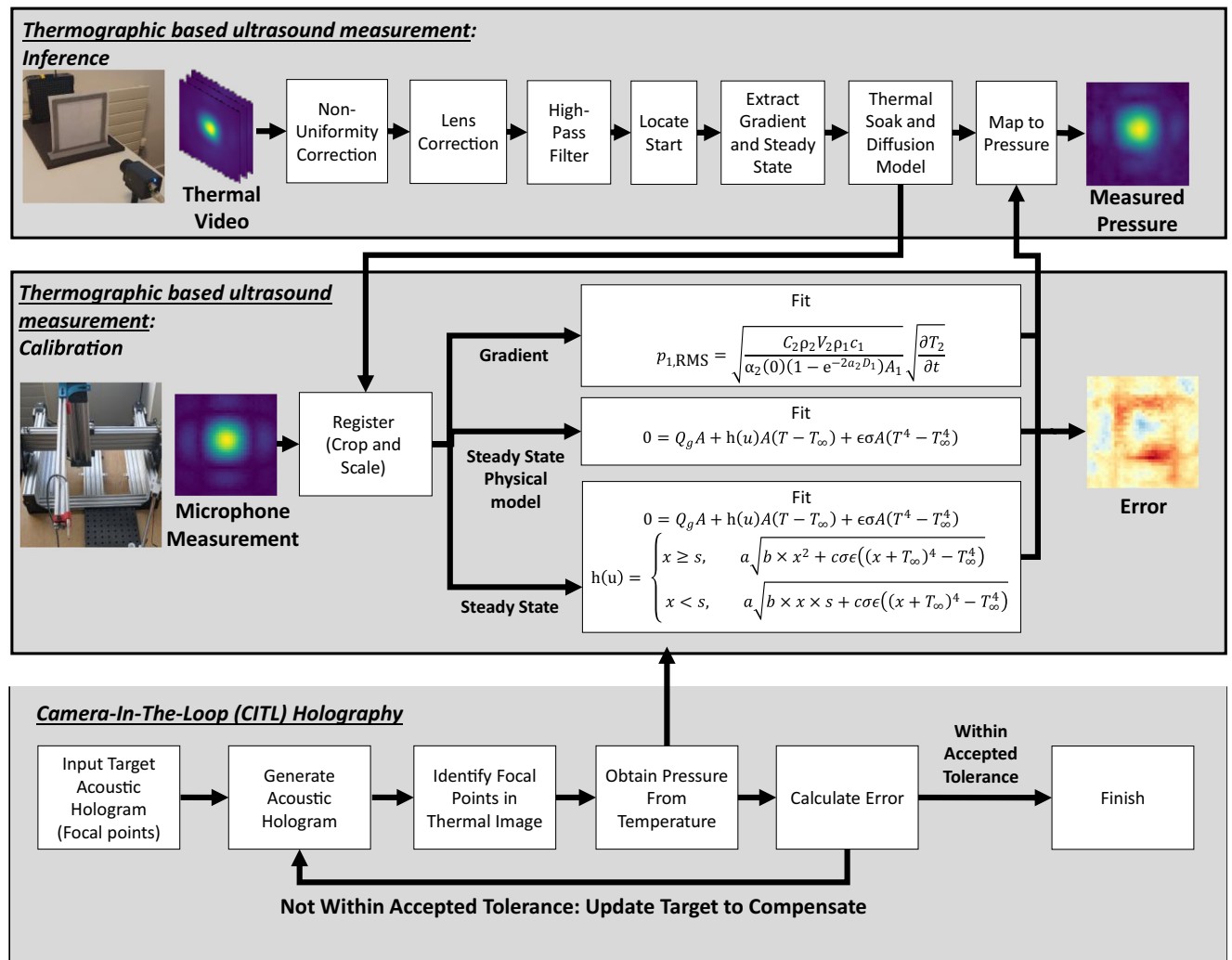

**Fig. 1 | Proposed measurement pipeline for sound fields utilising thermography and its integration with a camera-in-the-loop (CITL) holography technique.** The thermal set-up is shown including an acoustic source, nylon mesh to absorb sound, and a thermal camera to measure the heat generated from this absorption. A scanning microphone set-up is shown below it and was used to calibrate the technique. Three methods are shown including gradient, steady state, and steady with a physical model. These are based on the thermal gradient and steady state temperature, respectively with the physical model utilising the three-dimensional heat equation.

acoustic streaming, as with a ground-truth microphone measurement, the steady state solution's convective cooling can be solved for, which should represent the air velocity. This would in principle act like a 2D hot-wire anemometer[42]. This is important as streaming accounts for roughly 40% of acoustic radiation force above 600 RMS Pa[39].

## Results

### Physical thermo-acoustic models

In order to effectively absorb sound and transform it into heat energy, material choice is crucial. With that in mind we first look at models which we can use to find a good absorber, and also be utilised to map between temperature and pressure. We choose a model shown based upon modelling the measurement material as a MPP made up of small rectangular ducts. This is the perfect model for our case, as a woven mesh can be used and approximated as a perforated panel, and with a thin material thickness and small pore size, our material can easily be seen as an MPP. The MPP model covers narrowband absorption, perfect for our case as the transducers have a sharply tuned resonance[9], thus perfect for mono-tonal holograms.

We can then use known equations for the impedance of an MPP to determine the absorption coefficient $\alpha_2(\theta)$[38], and the low-frequency analytical models of a rectangular duct[43] for the attenuation coefficient $\alpha_2$. These quantities can then be used to determine the energy required to heat the material by a measured amount in the given time, and then calculate the acoustic pressure that would have to be present to dissipate this much energy in the material (Derivation in Supplementary Note 1 in Eq. (1) through 5):

$$p_{1,\mathrm{RMS}} = \sqrt{\frac{C_2 \rho_2 V_2 \rho_1 c_1}{\alpha_2(0)(1 - e^{-2a_2 D_1})A_1}}\sqrt{\frac{\partial T_2}{\partial t}} \tag{1}$$

Here subscripts entail the medium, where for example $\rho_1$ is the density of air, whilst $\rho_2$ is the density of the nylon mesh. $A$ and $V$ are surface area and volume respectively, $p$ pressure, $C$ specific heat capacity, $c$ the speed of sound, $D$ the thickness of the mesh and $T$ the temperature. The intuitive explanation of this formula is that we know how much power it takes to produce a certain rate of temperature change in a material by its specific heat capacity $C$ and mass (volume $V$ times density $\rho$). From this we can find out the heat flux by dividing power by area $A$, which must be equal to the acoustic flux, which is attenuated. Since we have a model for the proportion of sound transmitted ($\alpha_2$) and the attenuation $\alpha$, we can multiply attenuation by the distance travelled $D$ through the material, and work out to have attenuated that much flux, how much acoustic intensity was incident on the

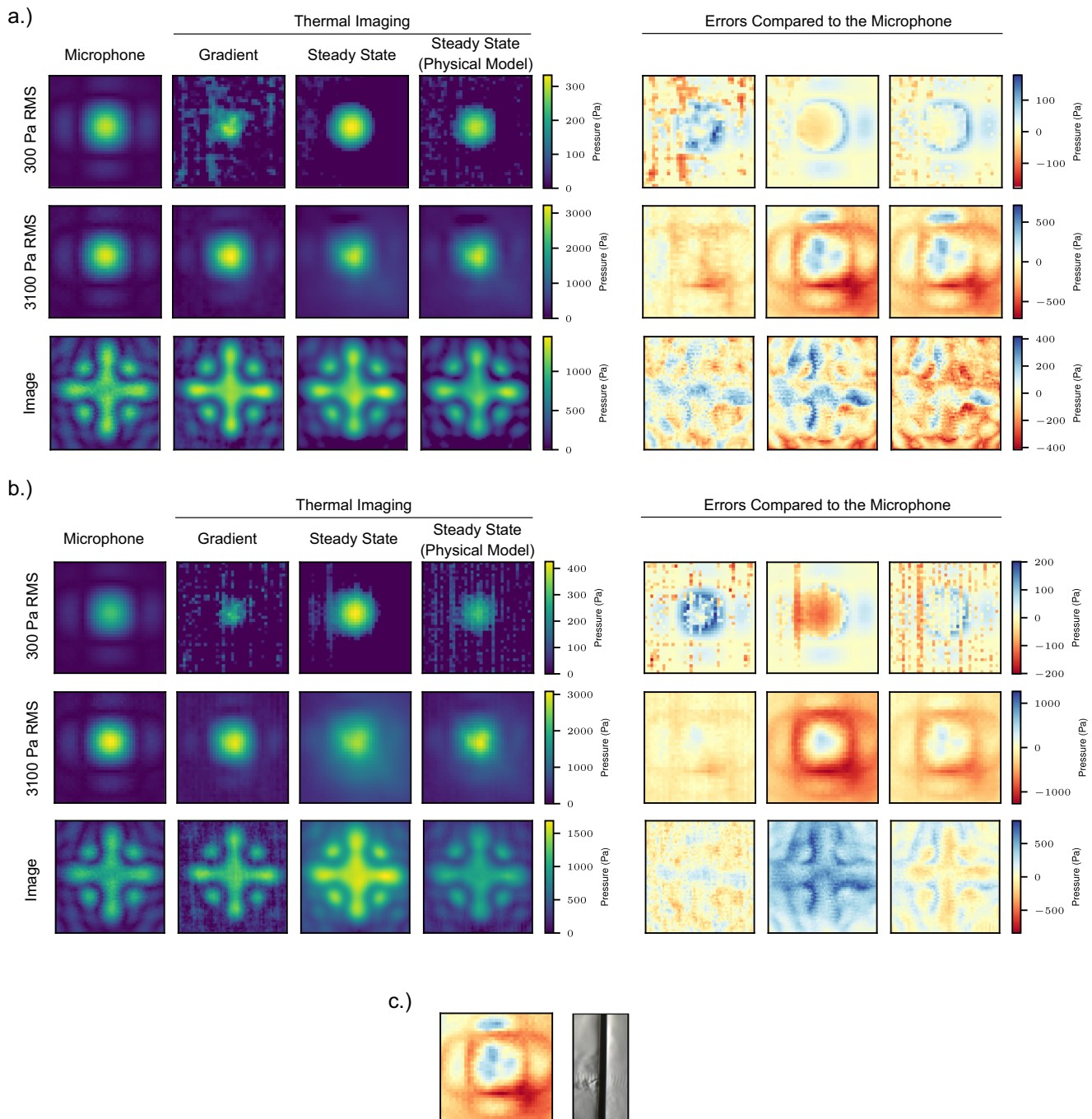

**Fig. 2 | Acoustic holograms measured by a microphone and by thermographic techniques with errors relative to the microphone data.** Two acoustic focal points are shown at high and low pressures, respectively, demonstrating different problems at high and low pressure. That is heat diffusion, and signal-to-noise ratio and cooling capacity surpassing heating, respectively. The root mean square (RMS) pressures extracted from three different thermography techniques are shown with their errors compared to a microphone scan and shown (**a**) with processing (noise reduction, heat modelling etc.) and (**b**) without any processing. **c** Highlighting the non-uniform error biased in the bottom right corner of a given focal point. The schlieren image clearly shows a corresponding high pressure air jet penetrating the mesh whose direction is caused by the particular weave pattern of the mesh.

material, and thus how much pressure was incident. This formula contrasts to that presented in ref. 37 which ignores the measurement material entirely, and assumes the temperature of the mesh and air are one and the same.

Using these models, the optimal material can also be chosen. To minimise reflections, a pore size of the same thickness as the viscous boundary layer should be chosen, and thus for 40 khz ultrasound about 23 μm, similar to the 25 μm we have chosen and demonstrated works well. Other sizes were tested and smaller sizes did indeed result in more reflection, whilst larger sizes resulted in too small an attenuation (See Supplementary

Figs. 1, 2 and Table 1). This mesh was irradiated with ultrasound from a phased array in a combination of scenarios with one or two focal points at various pressures ranging from 0 to 3000 Pa RMS. This mesh was imaged with a thermal camera and Fig. 3 shows the resulting data extracted from it. In this figure we show the lines of best-fit using various models which map temperature to pressure. We compare this model to the experimental data and find good agreement with an analytical absorption coefficient of 0.924 and attenuation of 835 Npm$^{-1}$, with the experimental data showing a best fit of 858 Npm$^{-1}$ for the given absorption coefficient. The absorption and

**Fig. 3 | Sound root mean square (RMS) pressure against thermal metrics in a mesh as well as the measured incident angle gain factor.** Pressure measured for a variety of single and dual focal point cases by a microphone and their peaks against (**a**) Initial temperature gradient of the nylon mesh showing the best fit, physical model simulation, and a model of the air temperature gradient[37]. **b** Steady state temperature reached after 10 s of the nylon mesh showing a hydro-acoustics model[31] and a model including both emission and convective cooling changes because of acoustic streaming. **c** Steady state physical model, showing estimated heat generation rate against pressure. Note this is not the measured gradient, but the estimated heat generation rate extracted from the steady state temperature. **d** Gain value compared to 0 degrees (perpendicular to the direction of propagation) at various incident angles for the microphone and thermographic techniques.

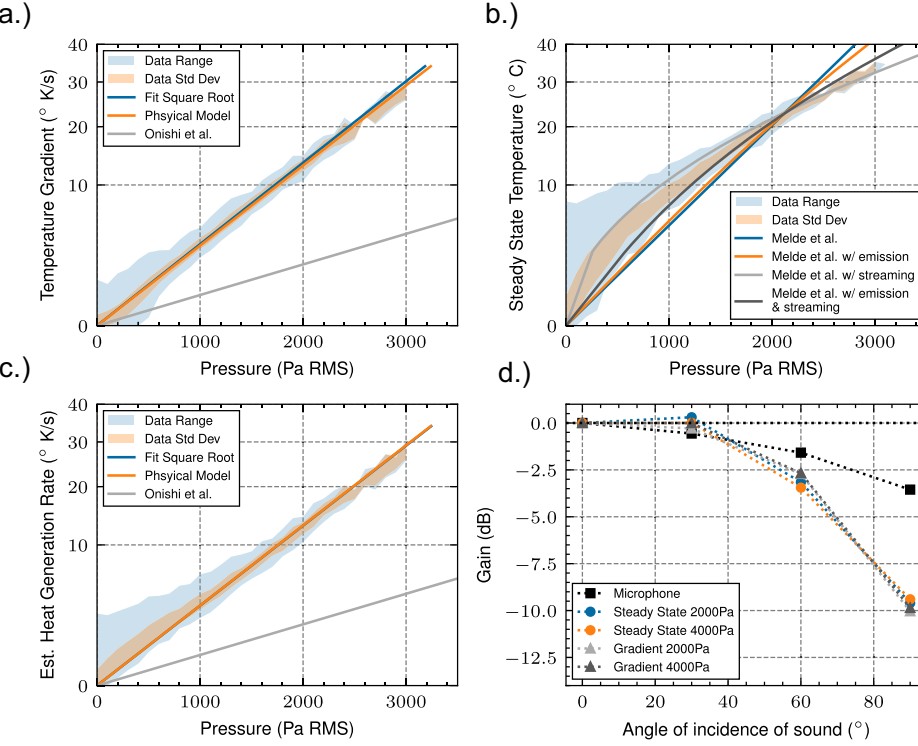

attenuation coefficients of the material were unable to be measured in practice, due to the small diameter required for an impedance tubes at such high frequencies.

Whilst a high-model accuracy is achieved over our data range here, the model has some limitations, notably due to its narrowband absorption spectrum, limiting non mono-tonal measurement[44]. Therefore, high energy harmonics due to non-linear effects are not well measured using this method, thus limiting accuracy in non-linear regimes. As in ref. 9 to the best of our knowledge there are no analytic equations to predicting acoustic saturation and thus non-linear effects in air. Due to this we defer to the data in ref. 9 which attempts to measure non-linear effects for devices like that used in this study, showing that when focussing at the distance we used (150 mm), saturation occurs experimentally at 165 dB. At our maximum measured pressure of 163.5 dB, it is therefore expected that we will see some non-linear effects. In Fig. 3 this is evident as the data starts to dip below the model at higher pressures.

The steady state data from Fig. 3 shows that the relationship is not the simple square root relationship as in water[31], but instead we must account for both emissive cooling losses, and streaming causing a changing convection coefficient. This results in a square root relationship, that transitions into a linear relationship. Streaming is further supported as the reason for this behaviour as it occurs around 500 Pa RMS, similar to the value found in ref. 39 and streaming velocity is approximately proportional to pressure[34], thus explaining the linear relationship.

For non-peak data-points there is a non-uniform and large spread in temperatures, presenting in a large increase in the data range for lower pressures. This makes creating a mapping between temperature and pressure difficult. This is due to the heat spreading out in the material as seen by the high error immediately surrounding the focal point in Fig. 2, and so limits the accuracy of any 1D calibration technique. This shows that whilst it is a valid technique for measuring the peak pressure of a focal point in real-time, it requires compensation for this heat spread in order to measure the area surrounding the focal point. In order to address this problem, we introduce two methods, a naive Gaussian heat spread model, and a heat-equation based physical model (PM).

The heat equation based PM allows for modelling conduction and convection that is not just based on the local steady state value, but also its

neighbours. The Gaussian heat spread model instead approximates heat as diffusing out in a Gaussian-shaped pattern around high pressure areas, such as the focal point. This is based on the fact that the fundamental solution to the heat equation is a Gaussian function[45]. The heat equation formula used for the PM is:

$$\rho C \sqrt{\frac{\partial T}{\partial t}} = kV\Delta T + Q_g A + h(u)A(T - T_\infty) + \epsilon \sigma A(T^4 - T_\infty^4) \quad (2)$$

Here $k$ is the thermal conductivity, $T_\infty$ the temperature of the environment, $h(u)$ the convection coefficient as a function of air speed, $\epsilon$ the emissivity, $\sigma$ the Stefan-Boltzmann constant and $Q_g$ the inner generation component, here attributed to the sound absorption and attenuation. The intuitive explanation of this equation is that the total heat energy change in our system is due to the summation of heat generated by acoustic attenuation, heat spread due to conduction within the material, heat lost or gained due to conduction with the ambient air via convection, and heat lost through emissive radiation.

In the steady state case, we have no temperature change $\frac{\partial T}{\partial t} = 0$, and so the right hand side of the equation must be equal to 0, allowing us to solve for our heat generation component $Q_g$ and thus pressure. The main difficulty in solving this equation is that pressure does not just effect $Q_g$, but also $u$ the air speed and thus $h$ the heat transfer coefficient. This equation also indicates that we are only in a steady state when our heat generation and our cooling are equal, an intuitive fact. This means the method is only valid when enough heat is generated to over-come the innate cooling of the material, thus setting a limit on the lower bound of pressure measurable. This does not depend on ambient conditions much, as it is the absolute temperature difference between ambient and the material which matters, and humidity only matters at higher than typical ambient temperatures, for example 100% relative humidity (RH) at 25 degrees Celsius is only a 3.6% increase in specific heat capacity compared to 0% RH.

It should also be noted that using the relationship from Figs. 3 and 4 from ref. 31, where pressure squared is proportional to pressure, with emission and streaming is equivalent to simply ignoring the conduction

**Fig. 4 | Pressures of each focal point in a 16 focal point hologram for GS-PAT and our camera in the loop (CITL) method.** Measured pressure via a microphone for a hologram of 16 focal points each at a 800 Pa (within device capabilities) and 1100 Pa (outside of device capabilities) target using GS-PAT with and without our CITL method. The results show a significant reduction in standard deviation for the 800 pa CITL method w/ soak, and a significant increase in accuracy for the 1100 Pa case. It is also noted that without accounting for the material temperature increase over time (soak), the method struggles.

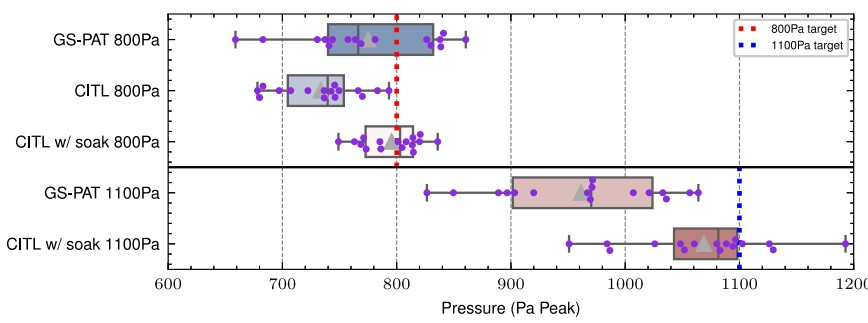

term such that:

$$0 = Q_g A + \mathrm{h}(u)A(T - T_\infty) + \epsilon\sigma A(T^4 - T_\infty^4) \qquad (3)$$

This gives us an equation that ignores any heat transfer internally in the mesh, and thus can be applied to each pixel in a thermal image independently of each other. If the temperature change is above some cut-off $s$ where we dictate streaming to start occurring, then we incorporate the convection coefficient being linked to air speed and thus pressure as per refs. 34,36:

$$\mathrm{h}(u) = \begin{cases} x \geq s, & a\sqrt{b \times x^2 + c\sigma\epsilon((x + T_\infty)^4 - T_\infty^4)} \\ x < s, & a\sqrt{b \times x \times s + c\sigma\epsilon((x + T_\infty)^4 - T_\infty^4)} \end{cases} \qquad (4)$$

In the PM, we incorporate conduction into the model, resulting in there no longer being a simple mapping between temperature and pressure. Instead in Fig. 3 we choose to transform the measurements using the heat equation to get energy, and then transform this into the initial temperature gradient that would result from that energy being incident on the mesh as in the gradient method. This allows us to use our models from the gradient method for calibration with the steady state PM. We can see that this method produces a much smaller range of data than the naive steady state method, and thus a lower error. As an approximation for the distribution of convection coefficient we can use the pressure mapped from the naive steady state method, given the assumption that streaming velocity is approximately proportional to pressure[34]. To sum up how this helps address our main problems, h(u) is now spatially dependent and addresses non-homogenous cooling, whilst $\Delta T$ addresses heat diffusion.

Since we have two unknowns in our steady state equation, heat generation due to pressure, and heat-convection coefficient, it is possible to instead solve for heat-convection coefficient if we have a known pressure. This could allow this method to instead be used to measure the air-velocity in acoustic streaming acting as a 2D hot-wire anemometer[42], though the impact on turbulence of this measurement would likely require further work.

Whilst ambient temperature $T_\infty$ was measured during the studies and used in the various formulas, variables which may vary based on it such as the speed of sound, density, viscosity and thermal conductivity were not adjusted as it was deemed to have only a minor impact on the results when compared to existing limitations and errors in the methods. For example, a change from 20 to 30 degrees Celsius in ambient does not change the specific heat capacity to the third decimal place, and only changes the speed of sound by about 2%. There is therefore the potential to increase accuracy further by modelling all these variables as functions of ambient conditions at the cost of increased complexity and data requirements.

## Thermographic measurement of acoustic holograms

With the mappings between temperature metrics and pressure worked out it is time to fully study their accuracy, as well as the methods for addressing things like noise, heat diffusion and acoustic streaming. Across the data set there are 18 full measurements with 10 having a resolution of $37 \times 37$ and 8 having a resolution of $55 \times 37$, resulting in 29,970 individual samples over the 18 test holograms. Table 1 shows the full results of the gradient, steady state and steady PM methods. We show the best results with processing and the raw results without processing, in order to demonstrate the effectiveness of the different methods and models. Full ablation results are shown in Supplementary Table 2. It is clear that the gradient method has the best results in all error types. This is likely down to the fact that the gradient method's errors are more uniform across the pressure range (Fig. 3), and thus the errors are likely down to noise, and the inaccuracy of using an uncooled microbolometer based thermal camera. This can further be seen in Fig. 2 in which it can be seen that the error is evenly spatially distributed, and also highlights the issues the gradient method has at measuring low pressures, where the signal-to-noise ratio (SNR) is low. It also clearly benefits greatly from noise reduction, resulting in the mean RMSE being reduced by about a third, and a similar decrease for FWHM error as well.

Noise reduction comprises of two methods. The first tackles the fixed-pattern noise (FPN) that is inherent to thermal imaging due to non-uniform sensor response using a Non-Uniform Correction (NUC). The second method applies a high-pass filter using a box-blur, applicable here as the fields to be measured contain only low-frequency features. In addition a non-uniform time sampling (NUTS) method is used where the higher the temperature, the less time-points are sampled to measure the gradient, thus limiting errors that occur at higher temperatures from the less accurate numerical differentiation (See Supplementary Fig. 3). Table 2 breaks out the performance impact of the NUC and blur. The $5 \times 5$ blur shows a 2% decrease in mean RMSE over the non-noise corrected case, and the NUC demonstrates a 22% decrease similarly, however, when combined we get a 28% decrease, showing that the two methods complement each other well. Further, we compare different blur sizes, with a square kernel size of 3, 5 and 7. The $5 \times 5$ kernel out-performs the $3 \times 3$ kernel on all metrics, whilst the $7 \times 7$ beats out the $5 \times 5$ kernel on all metrics apart from FWHM where we start to see a regression. This is to be expected as larger kernels will decrease the noise, but also start to impact the signal as they get bigger, especially in spatial measures of accuracy like FWHM or structural similarity index measure (SSIM).

The steady state methods perform on-par with each other in mean RMSE error and SSIM, however when it comes to the mean FWHM error we can see the physical method has a lead, likely due to the reduced error in the low-pressure regions as visible in Fig. 3. It does however require the processing, as without it the performance degradation is large, unlike the naive steady state model. Never the less both methods report results not too dissimilar to the gradient method. With the non-physical steady state method it is clear that the most important thing is to model convection, in order to capture the effect streaming has on cooling, which results in a 50% decrease in FWHM error, and a 10–15% decrease in RMSE. Modelling heat

**Table 1 | Thermal pressure measurement methods and their errors against microphone data, without any processing and with the processing that resulted in the lowest errors**

| Method | Model | Soak | Gauss | Noise reduction | Mean RMSE (Pa) | Mean max error (Pa) | Mean FWHM error (mm) | Mean SSIM |
|---|---|---|---|---|---|---|---|---|
| Gradient | NUTS | False | False | True | 64 | 227 | 0.37 | 0.59 |
| Steady State | 31 w/ emission, convection | True | True | True | 79 | 257 | 0.69 | 0.60 |
| Steady State (PM) | Gradient | True | True | True | 79 | 276 | 0.41 | 0.56 |
| Steady State (PM) | Steady state | True | True | True | 81 | 276 | 0.43 | 0.56 |
| Steady State | 31 w/ convection | False | True | True | 81 | 276 | 1.28 | 0.60 |
| Steady State (PM) | 35 image | True | True | True | 81 | 282 | 0.41 | 0.56 |
| Steady State (PM) | 35 | True | True | True | 82 | 283 | 0.43 | 0.56 |
| Steady State (PM) | Uniform | True | True | True | 94 | 306 | 0.83 | 0.54 |
| Steady State | 31 w/ emission | True | True | True | 220 | 532 | 2.37 | 0.47 |
| Steady State | 31 | True | True | True | 244 | 578 | 2.76 | 0.45 |
| Gradient | Raw | False | False | False | 89 | 284 | 0.51 | 0.49 |
| Steady State | 31 w/ convection | False | False | False | 94 | 312 | 0.97 | 0.55 |
| Steady State | 31 w/ emission, convection | False | False | False | 114 | 322 | 0.62 | 0.52 |
| Steady State (PM) | Gradient | False | False | False | 141 | 380 | 1.00 | 0.48 |
| Steady State (PM) | Steady state | False | False | False | 144 | 396 | 0.96 | 0.47 |
| Steady State (PM) | 35 image | False | False | False | 149 | 399 | 1.17 | 0.47 |
| Steady State (PM) | 35 | False | False | False | 153 | 403 | 1.14 | 0.47 |
| Steady State (PM) | Uniform | False | False | False | 167 | 418 | 2.18 | 0.44 |
| Steady State | 31 w/ emission | False | False | False | 303 | 615 | 3.49 | 0.39 |
| Steady State | 31 | False | False | False | 332 | 660 | 3.95 | 0.37 |

Lowest Mean root mean square error (RMSE) for each Method and model, and the raw results without processing and their respective mean RMSE, max error, FWHM (Full Width at Half Maximum) error and SSIM (Structural Similarity Index Measure), ordered by Mean RMSE. Model determines the sampling method for the gradient technique, the types of cooling considered for the steady state, and the convection coefficient distribution used for the PM. "Soak" determines if the temperature increase in the material not under stimulus was accounted for, "Gauss" if a Gaussian model was used to subtract estimated heat diffusion and "Noise Reduction" if a box-blur and static NUC (Non-Uniformity Correction) was used to reduce thermal image noise. (PM) stands for physical model.

**Table 2 | Noise reduction parameters and their errors sorted by mean root mean squared error (RMSE) for the gradient method**

| Method | NUC | Blur kernel size | Mean RMSE (Pa) | Mean max error (Pa) | Mean FWHM error (mm) | Mean SSIM |
|---|---|---|---|---|---|---|
| Gradient w/ NUTS | True | 7 × 7 | 63 | 218 | 0.38 | 0.60 |
| Gradient w/ NUTS | True | 5 × 5 | 64 | 227 | 0.37 | 0.59 |
| Gradient w/ NUTS | True | 3 × 3 | 67 | 237 | 0.40 | 0.58 |
| Gradient w/ NUTS | True | None | 69 | 250 | 0.44 | 0.57 |
| Gradient w/ NUTS | False | 5 × 5 | 87 | 268 | 0.51 | 0.51 |
| Gradient w/ NUTS | False | None | 89 | 285 | 0.55 | 0.49 |

Errors listed include the mean max error, FWHM (Full Width at Half Maximum) error and SSIM (Structural Similarity Index Measure). It is evident that the Non-Uniformity Correction (NUC) and the box-blur complement each other, and that the kernel size of the blur decreases noise, until it starts to smooth out the image too much at which point the FWHM error starts to increase.

diffusion via a Gaussian was also seen to be important, reducing FWHM error by 50–60% and RMSE by 30%.

In terms of predictive error for peak pressures, above 400 Pa RMS, the gradient method had an average error of 29 Pa and 2.5% and a max error of 121 and 8.0%, whilst the steady state method was 46 Pa, 3.5% and 107 Pa, 8.6%, respectively. Both of these values are better than the 150 Pa, 13% error reported in ref. 37 in air using the gradient method, and the 10 and 6% error in peak pressure for the two focal points in ref. 31 in water using the steady state.

The increase in accuracy over[31] is attributed to the much higher temperatures reached, due to the diminished cooling capabilities of air, compared to water, and thus an increase in the SNR. The increase in accuracy over[37] is attributed to 3 main factors. The primary reason being the measurement material being perpendicular to the direction of sound propagation, thus forcing the sound through the porous mesh, increasing attenuation and absorption and thus the gradient, thus increasing the SNR. The second and third reason are choosing a mesh with an optimal pore size for maximum absorption, as well as the noise-reduction techniques employed. These differences can be seen in Fig. 3 showing that the maximum temperature gradient reached is a little over 30 degrees per second for our method, in comparison to a little under 5 degrees per second for the method utilised in ref. 37, and additionally we utilise tens of thousands of data points to check accuracy, instead of only 1.

Despite the reduced accuracy of the steady state method over the gradient method, the steady state technique harbours a key advantage over the gradient method in successive measurements. This is that it can effectively measure in real-time, only requiring the material to cool to the new peak temperature in-between measurements. The gradient method however requires the material to cool to ambient between each measurement. This enables a CITL system utilising the steady state method. We can quantify this time as the gradient method takes approximately 1/3 of a second to measure, with the steady state measurement 10 s compared to 33 min (1 sample per second) for the microphone for the 37 × 55 1 mm resolution scan of the dual focal point data in this paper. In order to measure 3D volumes or successive measurements our thermographic methods however require a cooling period in-between measurements, with double the exposure time found to be sufficient to return to ambient. This makes the scan time for a volume $e(3n-2)$ where $e$ is exposure time and $n$ is the number of planes to scan for the gradient method.

This time taken is relevant not only for efficient measurement, but because that while it is a common assumption that the pressure field being measured is static, this is often not true. This is especially true with phased array of transducers (PAT) boards as their large amounts of transducers means they often generate a large amount of heat, and temperature changes the propagation of waves[46], therefore capturing quickly is paramount to avoid these effects. If the methods are to be used for real-time use, it is necessary that the computational costs also be low as well as the capture time of the data and thankfully the computational costs of these methods are cheap. The steady state method simply requires an image subtraction, Gaussian blur and NUC. With so much priority in fast image and video processing in software and hardware these are all achievable in real-time, with complexity depending on resolution, but for those used here this is sub-

millisecond. The gradient method is more computationally expensive as it requires multiple successive frames and linear-regression to fit the gradient. For the data used here time was 23.5 ms for the naive gradient and 112 ms for the NUTS noise reduction method. Given the recording rate of 30 fps, and the requirement of between 6 and 10 frames for each pixel we take between 200 ms and 333 ms per capture. Our gradient method is therefore also real-time. This is scripted in Python (high-level, interpreted language) which could be sped up by utilising an optimising compiler either by using a compiled language or a compiler for Python such as Cython or Pypy instead of the interpreter used here. The PM has the same order of computational complexity as the normal steady state technique, thus making all techniques presented here real-time.

Finally, it is worth noting causes of systematic error within our results. The first is the registration error between the thermal video data and microphone measurements, spatial averaging, down-scaling and registration error of the actual set-up, in that the microphone may not have been perfectly measuring the same plane as the thermographic measurement. This is hard to control as the mesh is not rigid. A further source of error in this evaluation is the inherent error of the ground truth microphone measurements themselves, since the microphone's directivity response is not uniform and the measurements were not taken in an anechoic chamber. Finally, due to the previously noted point, the device is active for half an hour during microphone measurement, and so it is possible for the hologram to fluctuate widely over this time-period.

A more qualitative evaluation is therefore also performed by looking visually at the results and using the SSIM. Figure 2 shows the microphone and thermal measurements compared for a single and dual focal point at different pressures, as well as an image based hologram. It is easy to see that the thermal methods are optimal at higher pressures, struggling to represent low-pressure focal points and artefacts such as side-lobes. Additionally, whilst the numerical errors of both thermal methods are similar in magnitude, it is clear that the gradient method is superior due to the heat diffusion problem, which even with modelling still greatly limits the steady state method. This is because the error is non-uniform in the low-pressure regions as can be seen especially in the bottom-right of the focal point. This is due to the mesh structure itself, as when the mesh is rotated this error also rotates. Using schlieren imaging as seen in Fig. 2 confirms this is due to the re-direction of hot air through the mesh due to the specific geometry created by the weave pattern. These results clearly show that whilst the numerical errors between the methods are comparable, the structure of the steady state results are less similar to the the microphone result.

Finally, a non-focal point target was also evaluated against consisting of a cross with a focal point in each corner, as seen in Fig. 2 with the error results in Table 3. The same behaviour as in with single and dual focal points was noticed, with nothing much particularly changing.

Another common disadvantage of microphones is their low upper pressure limit. In order to test higher pressure for the thermographic techniques, two PAT boards were measured during a levitation scenario. Individually each board can produce a focal point of about 6kPa peak, so combined if in a linear system, should produce about 12 kPa peak. The pressure was measured at 10.5 kPa peak and 9.2 kPa peak by the steady state and gradient method, respectively. Since this is far above the calibration data

**Table 3 | Errors for a non-focal point hologram with and without processing for all thermal pressure measurement techniques**

| Method | Model | Soak | Gauss | Noise reduction | RMSE (Pa) | Max error (Pa) | SSIM |
|---|---|---|---|---|---|---|---|
| Gradient | NUTS | False | False | True | 90 | 312 | 0.85 |
| Steady State | 31 w/ emission, convection | True | True | True | 127 | 398 | 0.78 |
| Steady State (PM) | Gradient | True | True | True | 137 | 416 | 0.78 |
| Gradient | Raw | False | False | False | 117 | 353 | 0.75 |
| Steady State | 31 | False | False | False | 402 | 842 | 0.66 |
| Steady State (PM) | Uniform | False | False | False | 130 | 447 | 0.77 |

Each method is shown ordered by root mean square error (RMSE) with their max error and SSIM (Structural Similarity Index Measure). Model determines the sampling method for the gradient technique, the types of cooling considered for the steady state, and the convection coefficient distribution used for the PM. "Soak" determines if the temperature increase in the material not under stimulus was accounted for, "Gauss" if a Gaussian model was used to subtract estimated heat diffusion and "Noise Reduction" if a box-blur and static NUC (Non-Uniformity Correction) was used to reduce thermal image noise. (PM) stands for physical model.

(since the microphone can measure up to only 5 kPa accurately, and 7 kPa without damage), it is hard to quantify accuracy, however this aligns well with the linear estimate of 12 kPa, given losses are expected when operating far into the non-linear pressure range[9] and other hardware idiosyncrasies prevent perfect device coupling.

A key limitation of this technique at high pressures is that the measurement material will exhibit the same properties as a levitated particle and be pushed into a node if not kept taut, and thus it is difficult to measure the maximum pressure which is present at the anti-node. This could also explain the difference in measured pressures versus expected. One possible way of getting around this limitation is by measuring with the mesh at an angle. Thus the directional response of the thermographic methods was measured and is shown in Fig. 3.

Using these values measurements were taken at 30 and 90 degrees, in order to avoid the mesh being manipulated in the direction of sound propagation. The values achieved were however not consistent and ranged from 7 kPa to 12.7 kPa for the gradient and from 5.3 kPa to 10.5 kPa for the steady state method. The theoretical limit of this technique for the saturation method is the melting point of the nylon mesh used. Given a melting point of approximately 264 degrees Celsius, that would correspond to a pressure of approximately 30 kPa provided the relationship stays the same with streaming velocities staying proportional to pressure. The limit for the gradient method is not clear, but likely much higher. It is likely that in most situations the saturation pressure in air will be below these limits anyway[9].

## Camera-in-the-loop (CITL) holography

With our thermographic measurement technique validated, we can now use it in our CITL system. Our CITL method acts as a wrapper around a standard acoustic holography algorithm (GS-PAT[11] in this case) and uses the steady state temperature measurement technique due to it not requiring the material to start at ambient temperature like the gradient method. The pipeline can be seen in Fig. 1 and is intended to show how even a simple PD loop can produce state of the art results when integrating pressure measurement feedback.

This CITL algorithm is designed to control $n$ focal points and their pressure $P_n$ as opposed to an arbitrary target. The system is tested for $n = 1$, where accuracy is the key metric, as well as in the multiple focal point case $n = 16$, where accuracy and standard deviation is are the key metrics. A single focal point with target pressures of 500, 1000, 1500, 2000, 3000, 4000 and 5000 Pa was created using the CITL algorithm (See Supplementary Fig. 7). In the single focal point case the RMSE and mean error between predicted and actual pressure was 47 and 38 Pa RMS, respectively. The error between the target and actual pressure, excluding the 5kPa case outside the capabilities of the device, was 34 and 29 Pa RMS, respectively which works out to approximately 1.8% and 1.7% relative error. In comparison GS-PAT averaged 7.8% average error across the single focal point data points collected for validating the thermal methods, significantly higher than the CITL (Welch's $t$-test $(-3.3310) = 8.0702$, $p = 0.0101$) and is notably different compared to the theoretical 2% error achieved in standard angular spectrum method simulations[11].

The 16 focal point case used equal pressure points with a target peak pressure of 800 Pa and 1100 Pa (around the maximum estimated capability of the device), 1.5cm from each other in two lines with 8 cm separating the lines vertically. The results of the CITL method are compared to GS-PAT on its own as seen in Fig. 4. In the 800 Pa case the mean appears closer to the target for the CITL method, but is not significant (Welch's $t$ $(-1.2761) = 20.1346$, $p = 0.2164$), although its reduction in standard deviation is (Brown–Forsythe $F(1,30) = 8.451$, $p = 0.0068$). For the 1100Pa case the opposite is true, with the increased accuracy in mean being significant (Welch's $t$ $(-4.4917) = 29.0170$, $p = 0.0001$), whilst the change in standard deviation is not statistically significant (Brown–Forsythe $F(1,30) = 0.7974$, $p = 0.3789$).

The 800 Pa case is also shown without accounting for soak, that is the temperature increase in the material over-time. This shows how important this becomes as the measurement time becomes longer, as the environment, board and(/or) material heats up. Importantly it is clear that errors are much larger than would be expected from simulations, thus again highlighting the importance of measurements when working in this domain. From these results it is clear the CITL method can easily allow an online improvement in algorithmic accuracy, especially when operating close to the device limits.

## Discussion

In this work we introduced and evaluated three methodologies for measuring in-air ultrasound pressure thermographically: one based on the initial temperature gradient and the other two on the steady state temperature achieved. These methods are many orders of magnitude faster than a typical scanning microphone set-up, making them suitable for capturing large amounts of data, performing real-time measurement and accurately capturing un-stable sound fields. It is also thought they could be used to visualise or measure non-static sound fields such as those utilised in spatio-temporal modulated haptics which are difficult to measure with a microphone (ref. 40) or whilst levitating. An example of this is visible in Supplementary Movies 1 and 2 however, further work on this was deemed beyond the scope of this work. We have summarised the key findings in Table 4.

A key component of our work as laid out in the summary table is the merging of MPP and thin-duct models with the previous thermographic techniques. This allows optimisation of the measurement material in order to maximise absorption and minimise reflection for a selected frequency. We found good agreement between the model and our data. Alternatively, it could be optimised to be as transmissive as possible while retaining the lowest porosity, and thus opacity, in order to have transparent displays with mid-air haptic devices behind them[47].

One main consideration highlighted by our work is that the geometrical structure of the mesh is also key. The weave structure in the mesh used in this work caused a re-direction of the acoustic streaming jet, thus transferring heat to the mesh non-uniformly. This was confirmed with a schlieren imaging system.

In order to increase the accuracy of the thermal measurements, different processing methods were employed. Noise reduction was found to play a critical role in ensuring the accuracy of the gradient method, and heat

**Table 4 | Summary table of key findings: thermographic measurement of in-air ultrasound pressure**

| Category | Gradient method | Steady state method | CITL | Summary |
|---|---|---|---|---|
| Speed | Very Fast (orders of magnitude faster than microphones) | Very Fast (orders of magnitude faster than microphones) | Very Fast | Faster measurement enabling real-time & unstable field capture as well as online holography using multiple pressure measurements. |
| Accuracy | Superior accuracy, simpler pressure relationship, but worse than optical methods | Less accurate than gradient method, complex pressure relation | Demonstrated improvements over GSPAT in single and 16 focal point holograms | Gradient method outperforms steady state in accuracy; both are significantly improved with pre-processing or more complicated models. |
| Advantages | Fast, accurate, simple calibration, high pressure capable | Fast, continuous/real-time measurement, high pressure capable | Real-time control and feedback | Non-scanning, fast, and somewhat accurate measurement for mono-tonal ultrasound fields & with potential for measuring high pressures; thermography viable for in-air ultrasound measurement. |
| Limitations/ Challenges | Noise sensitive at low pressure, sample size trade-off, directional response, material movement at high pressure | Complex pressure relation, heat diffusion modelling, directional response, material movement at high pressure, mesh geometry sensitive | Inherits steady state limitations | Heat diffusion modelling required at high pressures, directional response, mesh weave bias and signal to noise ratio is poor at low pressures. |
| Novelty | MPP and thin-duct model for mesh optimization, accounts for mesh material composition, advanced processing (non-uniform time sampling (NUTS), non-uniform calibration (NUC)) | In-air application, accounting for non-homogenous cooling caused by streaming & heat diffusion | Demonstration of online holography utilising multiple simultaneous real-time pressure measurements | Novel application of the MPP and thin-duct models, in-air steady state method and advanced processing techniques for improving accuracy. |
| Best use cases | Accurate measurements, shorter measurement time, calibration ease | Continuous/real-time measurement, CITL systems | Complex acoustic field control, haptics, levitation, digital twins | Characterizing & calibrating in-air ultrasound devices such as acoustic levitators, haptics, in-air SONAR or parametric speakers; data generation for acoustic holography. |

diffusion modelling in the steady state method. Furthermore, a trade-off in sample size was noted for the gradient method, where larger samples improved the signal-to-noise ratio but reduced the accuracy of the numerical differentiation, thus a method (NUTS) was devised for choosing the number of samples based on the magnitude of temperature gradient extracted.

Whilst the gradient method demonstrated a simple relationship with pressure, the steady state method exhibited a more complicated relationship due to non-uniform cooling caused by acoustic streaming and heat diffusion in the air, material. Metrics such as RMSE, maximum error, and FWHM were used for evaluation and it was found that the gradient method consistently outperformed the steady state method in all aspects, however, it is limited in its use case for real-time measurement compared to the steady state method.

We attempted to measure high pressures above the upper end of the range of conventional microphones, however struggled as a result of the high pressure physically moving the measurement material itself. In order to address this challenge the directional response of thermographic methods was investigated, highlighting differences compared to microphones. Correcting for this directional response allows placing the material such that the forces on it are parallel to the mesh instead of perpendicular, thus minimising movement. The results were however, too in-consistent to be conclusive and validation was not possible due to the microphones upper pressure limit being too low. The theoretical limit to this technique is hypothesised to be that of the melting point of the material used and therefore in the region of 30 kPa provided the relationship stays the same at higher temperatures and pressures. For the gradient method the limit is unknown, and likely to be when the non-linear acoustic regime starts to dominate, although as noted above the saturation pressure is likely to be below these limits in most situations[9].

We further demonstrated the use of the steady state thermal method in a CITL system, demonstrating its ability to control multiple focal points and achieve fairly accurate pressure control, exceeding that of traditional algorithms. This approach holds promise for applications requiring precise acoustic field manipulation, for example controlling the perception of ultrasound haptics[48] or more stable control of particles in levitation. This approach could also be used in a digital twin model[49], in order to provide offline training for holography algorithms, or collect large labelled data-sets.

In conclusion, we presented valuable insights into the measurement of acoustic fields using thermography and micro-perforated meshes, using gradient and steady state methodologies. Compared to existing works that used the gradient method[37], we utilise the MPP model to find a more ideal mesh hole size, resulting in a higher ratio of sound absorbed compared to transmitted, whilst minimising reflections, thus giving a much improved signal to noise ratio and a higher measureable minimum pressure. We further use this model to map between temperature and pressure, considering the mesh geometry and material such as the specific heat capacity of the mesh, rather than assuming the mesh and air temperature rises are identical and ignoring thermo-viscous effects. Finally, we introduce further processing on the thermal imagery, including non-uniform calibration to minimise noise, as well as non-uniform time sampling to minimise error when doing linear regression to extract the gradient, by adjusting the time-window used depending on the magnitude of the signal, maximising the signal to noise ratio. We further bring the steady state temperature technique[31] to the in-air domain, as well as extending it to account for the non-homogenous cooling we find in-air due to acoustic streaming, and the greater spread in heat diffusion within the measurement material, due to the diminished cooling capacity of air compared to water.

While this work offers new advancements, there are some inherent limitations that warrant further investigation such as the difficulty of heat diffusion modelling, low signal to noise ratio in low pressures, the directional response, and non-linear acoustic regimes as well as crucially the difficulty of validating high pressure measurements, as noted in Table 4. Addressing these challenges will be crucial for developing even more robust and versatile thermographic acoustic field assessment techniques.

With a rising number of commercial companies offering or researching in-air ultrasound acoustic based devices—such as levitators (Neurotechnology Ultrasound Research, BOROSA Acoustic Levitation GmbH, Materials Development Inc.), haptics (ultraleap[50]), directional audio (focusonics, Sennheiser) and imaging (Robert Bosch GmbH[51])—characterising, measuring and calibrating these devices is of commercial interest. The technique in this work could be adopted by these industrial and commercial entities to improve their devices.

In order to make these techniques as accessible as possible to a wide-range of users, a GUI is provided in order to allow easy capture, conversion and saving of measurements using the developed techniques. It further allows further calibration and validation in order to adjust the conversion process (Please see Supplementary Figs. 4, 5 and 6 and Supplementary Movie 2).

Moving forward, work should focus on optimising the material using the criteria discovered in this paper and better thermal modelling techniques to improve the accuracy and sensitivity of these techniques, in addition to exploring the accuracy achievable with more accurate cooled thermal cameras. This technique should also be used to create large data sets of real-world data for training holography algorithms, also allowing haptic designers to more accurately control perception of designed stimuli. Exploring measurement techniques from moving haptic holograms would also be of interest to the haptic field, especially for spatio-temporal modulated techniques. Finally exploring pressures far into the non-linear regions beyond the capabilities of microphones is also of paramount interest, given the current limitations on measuring ultrasonic levitation conditions when large amounts of transducers are used.

## Methods

### Analytical models for absorption and attenuation of perforated mesh

In order to calculate the absorption coefficient, we use the MPP model to calculate the bulk materials specific acoustic impedance using the following equations[38]:

$$z = \frac{Z}{\sigma \rho c} = r + j \omega m \tag{5}$$

where

$$r = \frac{8\mu t}{\sigma \rho c a^2} k_r, \quad k_r = \sqrt{1 + \frac{k^2}{32}} + \frac{\sqrt{2}}{16} k \frac{a}{t} \tag{6}$$

$$\omega m = \frac{\omega t}{\sigma c} k_m, \quad k_m = 1 + \sqrt{1 + \frac{k^2}{2}} + 1.7 \frac{a}{t} \tag{7}$$

$$k = a \sqrt{\frac{\omega \rho}{\mu}} \tag{8}$$

Where $\sigma$ is porosity, $\rho$ is density, $\mu$ is dynamic viscosity, $c$ is the speed of sound, $t$ is the thickness of the MPP and $a$ is the radius of a pore. Using this impedance the overall reflection and transmission ratios can then therefore be calculated by:

$$\alpha(0) = \frac{4r}{(1 + r)^2 + (\omega m - \cot(\omega D/c))^2} \tag{9}$$

where $D$ is the depth of the cavity behind the MPP if present.

In order to obtain the attenuation coefficient, we use the square analytical duct expressions assuming a Knusden number of approximately 0 and thus continuum flow[43]:

$$\alpha = \text{Im} \left( \frac{\omega}{c} \sqrt{\frac{\gamma - (\gamma - 1)\Psi_{th}}{\Psi_v}} \right) \tag{10}$$

$$\Psi_j = k_j^2 \sum_{m=0}^{\infty} \left[ 2(\alpha_m m')^{-2} \left( 1 - \frac{\tan(\alpha_m W/2)}{\alpha_m W/2} \right) \right] \tag{11}$$

$$\alpha_m = \sqrt{k_j^2 - \left( \frac{2m'}{W} \right)}, \quad m' = (m + 0.5)\pi \tag{12}$$

$$k_v^2 = -i \omega \frac{\rho}{\mu}, \quad k_{th} = -i \omega \frac{\rho C_p}{k} \tag{13}$$

Where for a given medium $\omega$ is angular frequency, $c$ is the speed of sound, $gamma$ is the adiabatic index, $W$ is the width of the duct, $\rho$ is density, $\mu$ is the dynamic viscosity, $k$ is the conductivity and $C_p$ is the heat capacity at constant pressure.

The assumptions required for these expressions are that: the acoustic wavelength must be much larger than the boundary layer thickness, the cross section of the ducts must be much smaller than the acoustic wavelength, the cross section of the duct must be constant or at most slowly varying in the propagation direction and the length of the duct in the propagation direction should be larger than the boundary layer thickness. For our chosen material we satisfy all of these apart from that the duct must be constant. This is due to the short length of the duct which is made up of round threads, thus instead of a square duct we have a square duct where the majority of the depth is flanged at the openings which could in future work be resolved with a simple end correction[52].

### Pressure and thermal data measurement

The design of the hardware used for the PATs is freely available from the OpenMPD framework github page of[53], as well as the OpenMPD software used to control them. They consist of 256 40 kHz transducers (MA40S4S, Murata Electronics, Japan) in a 16 × 16 grid with a pitch of 10.5 mm, with an average output of 120 dB at 30 cm when driven with a 20 V square wave. Solutions for a single focal point and double (2 cm separation) focii at 15 cm distance were generated using GS-PAT[11] under the OpenMPD framework[53]. For the single focal point target pressures of 0, 177, 354, 530, 884, 1237, 1591, 1945, 2298, 2652, 3889 Pa (RMS) were used reaching an actual measured peak pressure of 3, 116, 302, 490, 885, 1244, 1511, 1823, 2074, 2334, 3081 Pa (RMS), respectively. For the dual focal point target pressures of 0, 177, 530, 884, 1237, 1591, 1945, 2298, 3889 Pa (RMS) were used with measured peak pairs of pressures of (3, 3), (158, 150), (566, 544), (921, 883), (1208, 1169), (1453, 1415), (1707, 1656), (1896, 1851), (2357, 2316) Pa (RMS).

Microphone measurements were done using a Brüel & Kjær 1/8-inch Pressure-field Microphone Type 4138 connected to a Brüel & Kjær 2670 externally polarised pre-amplifier at $1\, mvPa^{-1}$ and the output measured using a PicoScope 4262. The microphone was oriented parallel to the surface of the PAT in order to be used as a free-field microphone, with the required 0.8 dB of gain removed from the response in order to correct the response as directed by the manufacturer.

2D scans of the focal points were measured over an area of 37 × 37 and 37 × 55 with a 1 mm step size. A nylon mesh of 25 $\mu m$ hole size with a thickness of 55 $\mu m$ and thread radius of 33 $\mu m$ was inserted into the focal point. A FLIR A35 thermal camera, with a resolution of 320 × 256 pixels and a sensitivity of 0.05 degrees then imaged the mesh for 20 s, with a 10 s ultrasound exposure, in order to also record the cooling properties, as well as heating.

Device, ambient and the bottom-left central transducer temperature was also recorded for measurements, and were kept at the same level between microphone and thermal measurements, in order to account for

any amplitude differences caused by device temperature. Due to the long time duration of the microphone scan, the temperature at the end of the microphone scan was taken, and the device was brought up to this temperature for the thermal acoustic measurements, as the device approximately reached a steady state during the scan. The ambient temperature was used when calculating temperature increases, and for the steady state PM method. Humidity was not measured or controlled for during this study, as its impact on the results is estimated to be small.

The impulse response of the transducers used was measured to be about 1 ms, as previously found in ref. 54, drastically lower than the minimum frame-time of the thermal camera of 16 ms (at 60 fps) and so was not a factor in our measurements as we use multiple frames, so it will be an error factor on the order of a couple of percent at most.

In order to minimise registration error between the thermal and microphone measurements we use normalised mean shifted cross correlation at multiple scales to choose the best match.

In order to correct for the lens distortion from the 9mm lens used with the thermal camera, OpenCV[55] is used to calculate the transformation from a series of checker-board images. The checker-board was heated by illumination using a high-powered LED in order to make it visible under thermal imaging.

When measuring the 16 focal point case presented in the CITL seciton, care must be taken to account for incident angles of any pressure wave due to the non-central location of the hologram due to the non-omnidirectional response of the microphone. When non-central points were measured, the angle between each transducer and the microphone was calculated and an individual free-field correction applied per transducer per point measured, using linear interpolation between the angles supplied by the manufacturer. This was done in simulation to estimate the change in magnitude of each focal point and then applied as an overall correction to the measurements respectively.

### Thermal to pressure mapping calibration

In order to fit our models to our data, the Trust Region Reflective method for least squares optimization was used as implemented by the scipy[56] curve_fit function.

Given the distribution of our data is skewed towards low temperatures, due to the background data points, as well as the shape of our pressure distributions, we must weight our data to apply optimization. The method used was to bin the steady state and gradient values by their square root rounded to one decimal place. Square root is used as that is the proportionality relationship between pressure and our variables. The variance within these bins was then used as the uncertainty (sigma) in the optimization.

Without this weighting, over-fitting to low pressure values is done sacrificing considerable accuracy at the higher pressure values, which are more important as they make up the foreground of the acoustic hologram.

### Noise reduction techniques

In order to get rid of noise both a static NUC to get rid of FPN and high-pass filter to get rid of high-frequency noise was used. A $5 \times 5$ box-filter average is used as a high-pass filter. Since the the same amplifier is shared amongst columns, each column has its own unique error[57]. Whilst the camera used has a built in NUC, it must close its shutter to do so, thus not being able to continuously change its NUC whilst recording. Therefore a scene-based NUC is required, that can identify noise in a non-uniform image. To design the NUC we represent the non-static FPN by a gain ($\mathbf{a}$) and offset ($\mathbf{b}$) factor such that at frame $n$ the image $\mathbf{Y}(n)$ is represented as a linear function of the original noise-free image $\mathbf{X}(n)$ as shown in Eq. (14). Gain is initially close to zero and drifts upwards temporally, whilst offset (additive) error is large initially and only drifts slightly higher with time[58], thus we can focus only on offset assuming $\mathbf{a} = \mathbf{1}$, since our capture times are short.

$$\mathbf{Y}(n) = \mathbf{a} \odot \mathbf{X}(n) + \mathbf{b} \tag{14}$$

Here $\odot$ is used to represent element-wise multiplication. We use the standard format of matrix-vector addition-multiplication notation where $\mathbf{C} = \mathbf{A} + \mathbf{b} \Rightarrow C_{i,j} = A_{i,j} + b_j$ and $\mathbf{C} = \mathbf{b} \odot \mathbf{A} \Rightarrow C_{i,j} = b_j A_{i,j}$. Since our measured signal only takes up a small portion in the centre of the image frame, there are a large number of rows that differ only by noise (Eqs. (15) and (16)), and can be used as a noise profile. Assuming the noise is approximately evenly distributed around the true value, with no bias for higher or lower values, we can simply take the mean as the true value. Therefore, the difference between the mean of a column and the mean of the signal-free section of the image represents the noise (Eq. (17)) and can be subtracted (Eq. (18)).

$$\exists n \forall k \Sigma_{j=0}^n X_{j,k} = \Sigma_{j=0}^n X_{j,k+1} \tag{15}$$

$$\exists n \forall k (\Sigma_{j=0}^n Y_{j,k} \neq \Sigma_{j=0}^n Y_{j,k+1} \iff b_k \neq b_{k+1}) \tag{16}$$

$$\frac{\Sigma_{k=0}^m \Sigma_{j=0}^n Y_{j,k}}{mn} - \frac{\Sigma_{j=0}^n (Y_{j,k})}{n} \approx -b_k \tag{17}$$

$$X_{j,k} \approx Y_{j,k} + \frac{\Sigma_{k=0}^m \Sigma_{j=0}^n Y_{j,k}}{mn} - \frac{\Sigma_{j=0}^n (Y_{j,k})}{n} \tag{18}$$

### Non-uniform time sampling (NUTS)

Non-Uniform Time Sampling changes the amount of samples used to calculate the numerical gradient, depending on the magnitude of the gradient. This is useful as larger gradients come with larger acceleration for a heat-curve, and thus the larger the gradient, the less-accurate the numerical gradient is for a given number of samples. First the highest sample value is used, and the gradient calculated. Any value falling above a cutoff is recalculated using a lower sample value until the list of cutoff values is exhausted. Values for the cutoffs was chosen to be [1,5,15,25] and the samples to [10,9,8,7,6] empirically.

### FWHM measurement

In order to measure FWHM, the maximum point of a focal point is found in a measurement. A horizontal and vertical slice is then taken and a Gaussian is fit using least squares to them. Fitting a Gaussian is done in order to eliminate noise, and provide a good estimate of the exact FWHM without needing more complicated non-linear interpolation on low-resolution measurements. This is valid as a focal point's shape closely follows a Gaussian distribution.

The Gaussian uses the equation

$$\mathrm{f}(x) = c + A e^{-\frac{(x-\mu)^2}{2\sigma^2}} \tag{19}$$

Since we have a constant added, the FWHM is thus the distance between the negative and positive half maxima points, $x_- + x_+ = \mathrm{f}\left(\frac{A+c}{2}\right) + \mathrm{f}\left(-\frac{A+c}{2}\right)$ (Note $\mu$ is ignored as it does not effect width) :

$$\mathrm{FWHM} = 2\sigma \sqrt{2\ln\left(\frac{1}{2} - \frac{c}{2A}\right)} \tag{20}$$

### Convection co-efficient distribution models

For the PM of the steady state method, we demonstrate 5 different models for the convection coefficient distribution. The 3 simplest are based on the data itself and are uniform, which is a uniform distribution, steady state, which uses the steady state temperatures, and gradient, which uses the square root of the gradients. The final 2 use streaming speed measured by ref. 35, with the ref. 35 method using a Piecewise Cubic Hermite Interpolating Polynomial interpolation of the data, and ref. 35 image, which scales the data using the standard image scaling technique of a 2nd order spline. All of these distributions are normalised, and then fit with a singular scaling factor depending on max temperature.

## Camera-in-the-loop (CITL) holography implementation

The equation used for our simple control loop is as follows:

$$\mathbf{u}(t) = K_p \mathbf{e}(t) + K_d \frac{d\mathbf{e}(t)}{dt} \tag{21}$$

This update term $\mathbf{u}(t)$ is simply added to the previous input. Here $K_p$ and $K_d$ are the non-negative coefficients for the proportional and derivative terms, respectively, whilst $\mathbf{e}(t)$ represents the error at time $t$. This error can be defined as the difference between the desired pressures $\mathbf{r}(t)$ and the measured pressures $\mathbf{y}(t)$:

$$\mathbf{e}(t) = \mathbf{r}(t) - \mathbf{y}(t) \tag{22}$$

In order to compare $\mathbf{r}(t)$ and $\mathbf{y}(t)$, we pair up measured focal points to target focal points based on their physical location. This is done by finding the local maximums in the thermal image, which are separated by a minimum distance, in order to avoid finding multiple local maxima in one focal point.

In order to obtain $\mathbf{y}(t)$ we must convert our measured temperature into an estimated pressure value. Thus we can construct a formulation for it as follows:

$$\mathbf{y}(t) = P_t(\text{Max}_{\text{local}}(\mathbf{x}(t) - \mathbf{x}(t_0)) \times 0.04 - s(t)) \tag{23}$$

Here $P_t(x)$ represents our calibration function which converts temperature change into pressure, $x(t)$ represents a thermal image at time $t$ and $s(t)$ is the temperature increase of the material at time $t$ due to factors other than absorption of ultrasound, like heat diffusion. The multiplication by 0.04 is needed, as an increase of 1 in the thermal image represents a temperature increase of 0.04 °C. In practice the value $s(t)$ is difficult to solve for as the temperature of the material is not homogeneous, and thus the closest point to the measured point is chosen which should have zero temperature change. That is the closest point by Euclidean distance for which the pressure is below 100 Pa.

Here, $\text{Max}_{\text{local}}$ returns the $n$ largest local maximas separated by at least $d_{\text{min}}$ distance, where $n$ is the number of focal points to be created and $d_{\text{min}}$ is the minimum distance between them in pixels in the thermal image.

To define $P_t(x)$ we create a simple function from section "Discussion" on thermal measurement. The function used here to will be as follows:

$$P_t(x) = \begin{cases} x \geq 2.8, & 23.17\sqrt{8.64 \times x^2 + 30.18\sigma\epsilon((x + T_a)^4 - T_a^4)} \\ x < 2.8, & 23.17\sqrt{8.64 \times x \times 2.8 + 30.18\sigma\epsilon((x + T_a)^4 - T_a^4)} \end{cases} \tag{24}$$

where $\sigma$ is the Stefan-Boltzmann constant, $\epsilon$ is the resistivity and $T_a$ is the ambient temperature.

## GS-PAT implementation

The implementation of GS-PAT[11] provided by the OpenMPD framework is used[53]. Both this implementation and that in the supplementary material by the original paper have a small inconsistency with their papers and implement the normalisation in the B matrix as:

$$\frac{P_{t,z}}{|\sum_t^T P_{t,z}^2|}$$

where as in the paper it is presented as:

$$\frac{P_{t,z}}{|\sum_t^T P_{t,z}|^2}$$

This results in a small numerical difference, both implementations were tested and it was noted that the correction helped slightly reduce standard deviation between points and increase accuracy in simulated results.

## Data availability

The data in this work is freely available in order to reproduce this work and can be found at https://doi.org/10.5281/zenodo.15365360[59].

## Code availability

The code containing a GUI front-end for this work is available at https://github.com/PhysiologicAILab/PyThermalMic[59]. The code used to produce the analysis in this work and figures is available at https://doi.org/10.5281/zenodo.15365360[59].

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

## Acknowledgements

This work was supported by the Royal Academy of Engineering through their Chairs in Emerging Technology Program (CIET18/19), the EPSRC through their Prosperity Partnership Program (EP/V037846/1) and the European Union's Horizon 2020 research and innovation programme under grant agreement No 101017746, project Touchless. The authors thank Ms. Ana Marques for their help in making the schematics and animations for this work.

## Author contributions

Z.M., Y.C. and S.S. conceived the project. Experiments, simulations and data analysis were carried out by Z.M. Z.M. wrote the paper, with contributions from all authors.

## Competing interests

The authors declare no competing interests.
