## [Transparent Peer Review file · Communications Engineering]

Rapid in-air Ultrasound Holography Measurement and Camera-in-the-loop Generation using Thermography

Corresponding Author: Mr Zak Morgan

Version 0:

Reviewer comments:

Reviewer #1

(Remarks to the Author)

This manuscript introduces a 2D thermographic measurement technique and a camera-in-the-loop holography algorithm to address the challenges in measurement and crafting of precise acoustic holograms. While the enhancements achieved by this work are impressive, several claims lack robust validation. Additionally, I have the following concerns:

1. The authors state that the proposed method can measure acoustic streaming, which is indeed intriguing. However, more detailed explanations are needed to support this claim, particularly regarding the mechanism by which acoustic streaming is captured and quantified using the described thermographic measurement technique.
2. The thermal-acoustic analytical model for thin-ducts and porous materials is presented as a key component of this work. However, thermal diffusion is inherently difficult to control and predict in both spatial and temporal domains. It remains unclear how the established analytical model ensures accurate and rapid measurements under such conditions. Further clarification and validation of this model are recommended.
3. In line 075, the manuscript mentions, "a normal scanning microphone set-up takes approximately 1 second to measure per "pixel", ...". The term "pixel" should be explicitly defined to avoid potential misunderstandings among a broader audience. Readers unfamiliar with the acoustic measurement context may misconstrue this terminology.
4. The authors claim that their method is advantageous due to its "small size, simple material and device requirements, and the availability of off-the-shelf components, as well as its independence from computed tomography algorithms". However, this sentence is difficult to follow. More importantly, the specific advantages of the proposed method over existing techniques remain unclear. Comparative analysis with prior works should be included to substantiate this claim.
5. The current validation of the method is limited to acoustic holograms patterned with simple single points. It is recommended to extend the validation to more complex acoustic hologram patterns.
6. The manuscript lacks information on when a steady-state temperature can be reached during the measurements. Additionally, the potential variability in steady-state temperature under different environmental conditions should be addressed, as this may influence the measurement speed and accuracy.
7. The conclusion that "diminished convection cooling capacity, decrease in frequency, density, and sound speed, as well as significantly larger focal points, lead to heightened and diffuse temperatures and non-homogeneous cooling behavior due to acoustic streaming" (line 129) is inadequately supported. Further explanation and experimental evidence are required to substantiate this statement.
8. The claim that "acoustic streaming can lead to non-homogeneous cooling behavior" is intriguing, but supporting evidence is absent. Additionally, the manuscript should explain how the proposed method mitigates the influence of acoustic streaming to ensure reliable measurements.
9. While the Full-Width at Half-Maximum (FWHM) can be used to analyze the acoustic pressure, it is unclear how this parameter supports acoustic force analysis and validates the integration of force over the target area for applications such as haptics and large-object levitation. Additional justification or analysis is needed to establish this connection.
10. The manuscript mentions reducing errors caused by noise, heat diffusion, and non-homogeneous cooling due to acoustic streaming. However, the specific mechanisms or techniques employed to achieve this reduction are not detailed. Clarification and experimental evidence should be provided.
11. The purpose of validating acoustic focal points at both high and low pressures in Fig. 2 is unclear. The manuscript should explain why such validation is necessary and how it contributes to the overall findings.
12. Fig. 3(c) and its corresponding caption are confusing and appear similar to Fig. 3(a). The authors should provide a clear explanation of the differences and relevance of these figures to avoid misinterpretation.
13. The manuscript employs an analytical thermo-acoustic model based on a micro-perforated panel (MPP). However, the rationale for choosing this specific model is not explained. The authors should justify the suitability of the MPP model for their study.
14. The abbreviation of "rms" should be defined for clarity, and the variable D1 in equation (1) requires a clear explanation of its physical meaning.

15. The sentence in line 257, "The absorption and attenuation coefficients of the material were unable to be measured at the," is incomplete. Such errors should be thoroughly reviewed and corrected throughout the manuscript to ensure clarity and professionalism.

16. The statement in line 258, "The high model accuracy should not necessarily be assumed without testing different measurement materials," is vague. The authors should elaborate on why high accuracy should not be assumed and provide experimental evidence or reasoning to support it.

17. The thermographic measurement results presented in the manuscript are overly simplistic. To strengthen the findings, both quantitative results and perceptual validation should be included, providing a more comprehensive evaluation of the technique.

18. The manuscript lacks sufficient validation for the precision acoustic generation achieved by the proposed camera-in-the-loop holography algorithm, which is presented as a core contribution. Experimental demonstrations and quantitative analyses should be provided to substantiate its effectiveness.

19. The manuscript requires significant grammatical and structural improvements to enhance clarity, flow, and readability. A thorough review of the language and logical structure is recommended.

20. Several conclusions presented in the manuscript lack proper citations. To improve the scientific rigor and credibility of the work, the authors should add appropriate references to support their claims.

Reviewer #2

(Remarks to the Author)

In this work, the authors introduce a rapid 2D thermographic measurement technique, which is orders of magnitude faster than microphone scans, at the cost of acceptable losses in accuracy and phase information, with a maximum peak pressure of 12kPa measured, and a demonstrated average accuracy of 2.5% in peak measurement. The manuscript is well organized and the revision suggestions are as following.

It is suggested to address the problem raised in the first paragraph. One equation or one figure is expected for a better understanding of the principle of the holography in this work, to those who are not in this field. It is suggested to explain what are the hologram plane and the target plane, respectively. In optical holography, finding the exact solution of a desired hologram to reconstruct an accurate target object is an ill-posed inverse problem. Various non-convex optimization algorithms are thus designed to seek an optimal solution by introducing different constraints, frameworks, and initializations (Light: Science & Applications 13, 158 (2024)). Then it is suggested to explain why the Camera-In-The-Loop (CITL) Holography is used in this work. Thus, the motivation and the background of this work could be highlighted.

The real-time measurement and processing are important for the task. It is suggested to address the calculation time for the algorithms in this work. There should be a balance between the calculation speed and accuracy.

Reviewer #3

(Remarks to the Author)

Comments on the paper entitled, "Rapid in-air Ultrasound Holography Measurement and Camera-in-the-loop Generation using Thermography" by Zak Morgan et al.

General Comments:

The authors have presented a rapid 2D thermographic measurement technique that could be a valuable insight into the thermographic assessment of acoustic fields using gradient and steady-state methodologies. The author reported that the technique is faster than microphone scans but at the cost of acceptable losses in accuracy and phase information. Further, the technique is integrated with holography algorithms to establish a camera-in-the-loop algorithm that employs real-time measurement, enabling targeted data acquisition and on-line training of acoustic holography algorithms.

Overall, the manuscript is written and documented well.

Major Concerns:

There are several concerns listed below:

1. The major concern is the novelty of the proposed study compared to existing methods. The authors should discuss in detail the novelty and major findings.

2. Does the proposed study consider all the following conditions:

- non-linear effects,
- ambient and device conditions,
- such as temperature [8, 9],
- humidity,
- device properties, such as the impulse response of transducers.

3. The literature survey should be improved by citing the works on sound field imaging by digital holography, e.g., Prof. Awatsuji's group.

4. Detailed explanations of Fig. 1 and 2 should be provided in the text.
5. The computation time to get the data in each case should be provided.
6. How does noise reduction play an important role in ensuring the gradient method? Quantitative analysis should be discussed.
7. The authors have used a 7×7 box filter average is used. Is it the optimized size?

Reviewer #4

(Remarks to the Author)

This paper proposes a fast two-dimensional ultrasonic holographic measurement technology based on thermal imaging. The measurement speed is significantly improved by the thermal gradient method and the steady-state temperature method, and the camera closed-loop (CITL) algorithm is combined to achieve high-precision real-time control of the sound field. The research has important application value in the fields of ultrasonic tactile feedback and acoustic levitation. The theoretical model is closely combined with experimental verification, and the overall innovation is strong. However, the paper has certain deficiencies in experimental details, data analysis, theoretical model verification, etc., which need to be further supplemented and improved. The following are specific review comments:

Combined with the CITL algorithm, real-time control of the sound field is achieved, providing a new technical path for applications such as ultrasonic tactile feedback and acoustic levitation. However, the paper's discussion of technical limitations is relatively superficial, and the impact of material melting threshold, nonlinear acoustic effects, etc. on high-pressure measurement is not deeply analyzed.

The performance evaluation of the CITL algorithm is limited to single-point and multi-point focus, and it is recommended to expand it to more complex sound field distributions (such as dynamic sound fields or non-uniform sound fields).

The experimental design is generally reasonable, but there is a lack of clear explanation of the number of experimental repetitions and sample size. It is recommended to supplement the statistical significance analysis of the experiment (such as t-test or ANOVA).

The paper does not discuss in depth the limitations of thermal imaging technology in high-pressure measurement (such as material melting and nonlinear acoustic effects). It is recommended to add relevant experimental data or theoretical analysis.

The applicability of the CITL algorithm in complex sound fields is not discussed enough, and it is recommended to expand the experimental scenario to verify its robustness.

Reviewer #5

(Remarks to the Author)

Review Report

The manuscript presents a novel thermographic technique for rapid 2D measurement of in-air ultrasound holography, providing a significant improvement over conventional scanning microphone techniques. Additionally, the authors integrate this measurement approach with a camera-in-the-loop (CITL) holography algorithm to enhance real-time feedback and accuracy. The study is well-motivated, addressing a critical limitation in ultrasound-based haptics, volumetric displays, and acoustic holography. The results indicate orders-of-magnitude speed improvement while maintaining reasonable accuracy, making this technique relevant for real-world applications. Although I acknowledge the idea and efforts behind the work I do have some concerns which I have shared in the following section.

1. Scientific Merit and Novelty

Strengths:

- The thermographic measurement method is novel and presents a compelling alternative to microphone-based approaches.
- The integration of real-time measurement with CITL holography improves algorithm adaptability and accuracy.
- The study accounts for critical factors such as non-linearity, acoustic streaming, and heat diffusion.

Weaknesses:

- While the method shows promise, a more comprehensive comparison with other established techniques such as Schlieren imaging, Laser Doppler Vibrometry (LDV), and heterodyne interferometry is needed to better establish its advantages and trade-offs.
- The validation of pressure measurements beyond 10 kPa remains uncertain due to the limitations of the reference microphone system.

Observation 1: The manuscript has some novelty with high scientific merit but requires stronger comparative analysis.

2. Technical Rigor and Methodology

Strengths:

- The experimental validation is thorough, with comparisons to conventional microphone-based approaches.
- The manuscript provides detailed modeling of thermographic pressure measurements, incorporating material properties and wave interactions.
- Quantitative performance metrics (RMSE, peak error, FWHM) are used effectively to support the method's validity.

Weaknesses:

- Some aspects of the mathematical modeling could be presented with more intuitive explanations for accessibility to a

broader audience.

- The accuracy of measurements under extreme conditions (>10 kPa) is uncertain and requires additional validation.

Observation 2: The methodology is well-structured but would benefit from additional validation and clearer explanations of key modeling assumptions.

3. Impact and Suitability for Nature Communications Engineering

Strengths:

- The study introduces a promising breakthrough in ultrasound holography measurement, enabling fast and scalable data acquisition.
- Potential applications in mid-air haptics, volumetric displays, and non-contact acoustic measurement make the study impactful.
- The real-time adaptability of CITL for algorithmic improvements aligns with modern engineering trends.

Weaknesses:

- The paper should discuss more explicitly how this technique can be adopted in industrial and commercial settings to be in alignment with the scope of the Nature Communications Engineering journal

Observation 3: The impact is significant within its niche but may not fully meet the broad interdisciplinary scope expected by Nature Communications Engineering.

4. Clarity and Presentation

Strengths:

- The manuscript is well-structured with a logical flow from problem statement to solution.
- Figures and tables effectively illustrate the methodology and results.
- The inclusion of a GUI for measurement validation enhances accessibility.

Weaknesses:

- Some sections, particularly the analytical modeling, are mathematically dense and could benefit from additional intuitive explanations.
- A summary table highlighting key findings would improve readability.
- Experimental details regarding calibration and data reproducibility should be explicitly stated.

Observation 4: The manuscript is well-written but would benefit from improved clarity in technical explanations and structure.

Recommendation

After carefully evaluating the manuscript, I provide the following recommendation:

Major Revisions Required (Before Reconsideration for Communications Engineering)

Required Revisions:

1. Provide a stronger comparison with alternative measurement techniques (e.g., Schlieren imaging, LDV) to better position the method.
2. Clarify the accuracy and limitations of high-pressure measurements (>10 kPa) with additional validation.
3. Improve accessibility of mathematical modeling by adding intuitive explanations where possible.
4. Discuss the broader impact beyond ultrasound holography, emphasizing interdisciplinary relevance.
5. Enhance clarity in presentation by restructuring dense sections and adding a summary table of key findings.

Version 1:

Reviewer comments:

Reviewer #1

(Remarks to the Author)

The reviewer has carefully reviewed the revised manuscript and appreciate the authors' efforts in addressing the reviewer's comments. All the comments were adequately responded to. The reviewer would like to recommend acceptance of this work to Communications Engineering.

Reviewer #2

(Remarks to the Author)

All the comments have been properly considered in the revised manuscript.

Reviewer #3

(Remarks to the Author)

In the revised manuscript, the authors have made satisfactory amendments in response to the raised questions and concerns. In my opinion, the manuscript is suitable for publication.

Reviewer #4

(Remarks to the Author)

The authors have addressed my question.

Reviewer #5

(Remarks to the Author)

Thank you for your revised manuscript. I appreciate the efforts made in addressing the concerns raised in the previous

round. The authors have satisfactorily responded to the issues, and the revisions have improved the clarity and quality of the manuscript. I have no further comments. I wish you all the best with the publication of your work.

We would like to thank the editor and the reviewers for reviewing the manuscript and providing critical and crucially very helpful comments. We have carefully gone through the comments and have revised the manuscript accordingly. We believe that these changes significantly improve the manuscript.

The changes in the manuscript are highlighted in yellow and here we respond to each point raised by the reviewers in turn.

Reviewer 1:

This manuscript introduces a 2D thermographic measurement technique and a camera-in-the-loop holography algorithm to address the challenges in measurement and crafting of precise acoustic holograms. While the enhancements achieved by this work are impressive, several claims lack robust validation. Additionally, I have the following concerns:

1. The authors state that the proposed method can measure acoustic streaming, which is indeed intriguing. However, more detailed explanations are needed to support this claim, particularly regarding the mechanism by which acoustic streaming is captured and quantified using the described thermographic measurement technique.

We agree that the manuscript was vague about this claim and have explained the proposed method in that it would function effectively like a 2D hot-wire anemometer. (see lines 490-495)

From equation three:

$$0 = Q_g A + h(u) A (T - T_\infty) + \epsilon \sigma A (T^4 - T_\infty^4)$$

Instead of solving for Q_g by making assumptions on $h(u)$, we would instead solve for $h(u)$, and then further extract u from this. This could be done by measuring with a microphone to get Q_g for example, although it is noted that the impact of the amount of turbulence at any point would introduce difficulty in this method, but that is expected.

2. The thermal-acoustic analytical model for thin-ducts and porous materials is presented as a key component of this work. However, thermal diffusion is inherently difficult to control and predict in both spatial and temporal domains. It remains unclear how the established analytical model ensures accurate and rapid measurements under such conditions. Further clarification and validation of this model are recommended.

We think perhaps our manuscript has been unclear, but the analytical models for thin-ducts and porous materials are used to calculate the absorption and attenuation

co-efficients, but not to model thermal diffusion. This confusion may arise as we used the terms thermos-acoustic and thermos-viscous in the manuscript, with the latter being what we have just discussed and the former being our models for how we go from the acoustic equations to our chosen temperature metrics. We notice we made a mistake in the abstract confusing these terms and have fixed it accordingly.

3. In line 075, the manuscript mentions, “a normal scanning microphone set-up takes approximately 1 second to measure per “pixel”, ...” The term “pixel” should be explicitly defined to avoid potential misunderstandings among a broader audience. Readers unfamiliar with the acoustic measurement context may misconstrue this terminology.

We agree this could be confusing, and so we have added our definition of “pixel” to the manuscript in the introduction. (lines 89-94)

4. The authors claim that their method is advantageous due to its “small size, simple material and device requirements, and the availability of off-the-shelf components, as well as its independence from computed tomography algorithms”. However, this sentence is difficult to follow. More importantly, the specific advantages of the proposed method over existing techniques remain unclear. Comparative analysis with prior works should be included to substantiate this claim.

We have expanded upon our advantages stating that they are cost (a mesh can be bought for a few dollars and a cheap thermal camera can be had for around \$100-200), simplicity in terms of equipment pieces (no lenses, optical tables, beam splitters etc....), robust to alignment (vibrations for example can cause the beam to miss the knife edge in schlieren) as well as there being no need for computer tomography algorithms or phase-unwrapping. In regards to the two previous thermal works, the advantage is clear in figure 3, where we present our calibration curves against the existing ones showing the clear advantage of our techniques, as well as table 1 showing the increase in accuracy. We have expanded upon those in the manuscript in order to attempt to make this clear. We do note our methods downsides in comparison, for example that phase is not able to be measured, and the sensitivity and signal to noise ratio at low pressures is significantly worse.

Lines (106-133, 169-183, 869-886)

5. The current validation of the method is limited to acoustic holograms patterned with simple single points. It is recommended to extend the validation to more complex acoustic hologram patterns.

Our method does not just use single points, but also dual points which are separated by 2cm center to center (line 973-974). We do however agree and so have added the measurement of a more complicated acoustic hologram consisting of a cross with four points in each corner to demonstrate that the method holds up for more complicated patterns. (see figure 2 and lines 671-674)

6. The manuscript lacks information on when a steady-state temperature can be reached during the measurements. Additionally, the potential variability in steady-state temperature under different environmental conditions should be addressed, as this may influence the measurement speed and accuracy.

We have added information stating that the steady state temperature can be reached whenever the heat generation due to acoustic attenuation is greater than that due to heat transfer out of the mesh, or into other parts of the mesh. We have added some information about environmental conditions and their impact on the steady state temperature, noting that in normal room conditions, they are minimal at the temperature levels we measure in the work. (lines 449-462)

7. The conclusion that “diminished convection cooling capacity, decrease in frequency, density, and sound speed, as well as significantly larger focal points, lead to heightened and diffuse temperatures and non-homogeneous cooling behavior due to acoustic streaming” (line 129) is inadequately supported. Further explanation and experimental evidence are required to substantiate this statement.

We have expanded this statement to further explore these differences and the large impact they have on the work. We include citations and data from previous works calculating the impact for example that streaming would have on the amount of convective cooling. Lines (139-142)

8. The claim that “acoustic streaming can lead to non-homogeneous cooling behavior” is intriguing, but supporting evidence is absent. Additionally, the manuscript should explain how the proposed method mitigates the influence of acoustic streaming to ensure reliable measurements.

We have added supporting citations to this claim, explaining that it is well known that pressure and acoustic streaming velocity in air are linked, and that air velocity and heat convection co-efficient are also linked, thus through hypothetical syllogism logic, we can deduce that pressure and heat convection co-efficient are linked. We also a numerical example from data in the literature with a reported streaming velocity of 2.9 m/s for a 2kPa RMS focal point that the heat transfer co-efficient would rise from 5 to 20-45 $\text{\$Wm}^{-2}\text{K}^{-1}$ depending on the level of turbulence present. (Lines 140-145) In the paper we lay out that acoustic streaming does not occur fast enough to impact the gradient method, and that we attempt to account for it by modelling our heat transfer co-efficient as proportional to pressure above a cut-off where we estimate streaming to start, as evident in equation 4 of the steady state, or by $h(u)$ in the physical model. (473-476)

9. While the Full-Width at Half-Maximum (FWHM) can be used to analyze the acoustic pressure, it is unclear how this parameter supports acoustic force analysis and validates the integration of force over the target area for applications such as haptics and large-object levitation. Additional justification or analysis is needed to establish this connection.

Since force is pressure over area, it is evident that the spatial distribution of pressure impacts force, and so FWHM was chosen as a metric which can inform our accuracy of measuring force as well as pressure. This is supported by haptic works calculating acoustic force via the integration of a gaussian distribution parameterised by peak pressure and FWHM. In addition in levitation the Gor'kov potential is often used, which is related to the derivative of pressure, which again FWHM errors act as a good proxy, as steeper gradients lead to a smaller width and vice versa. We have added citations to these works that use these methods as supporting evidence in the manuscript.(Lines 220-225)

10. The manuscript mentions reducing errors caused by noise, heat diffusion, and non-homogeneous cooling due to acoustic streaming. However, the specific mechanisms or techniques employed to achieve this reduction are not detailed. Clarification and experimental evidence should be provided.

The specific mechanisms have been explained including high-pass filters, non-uniform corrections, non-uniform time sampling, gaussian heat models, solving the heat-equation to account for convection, conduction and emission as-well as various models for proportionality of pressure to cooling to account for streaming. In addition the ablation table provided demonstrates the effectiveness on error that these have, as-well as new tables for noise reduction.

11. The purpose of validating acoustic focal points at both high and low pressures in Fig. 2 is unclear. The manuscript should explain why such validation is necessary and how it contributes to the overall findings.

This is a good point and we have added the justification in the text explaining that at low temperatures, the signal to noise ratio is much lower, thus noise is the primary concern, whilst at high temperatures, heat diffusion and non-linear effects such as streaming are larger concerns. (lines 232-237).

12. Fig. 3(c) and its corresponding caption are confusing and appear similar to Fig. 3(a). The authors should provide a clear explanation of the differences and relevance of these figures to avoid misinterpretation.

We agree and have changed the axis on Fig. 3(c) and the caption in order to better differentiate them, as well as adding text in the main manuscript to avoid any confusion. (Lines 408-410, 477-481)

13. The manuscript employs an analytical thermo-acoustic model based on a micro-perforated panel (MPP). However, the rationale for choosing this specific model is not explained. The authors should justify the suitability of the MPP model for their study.

The justification was indeed absent and so we have added reasoning behind this decision which is the following: A mono-tonal hologram is to be measured, and so a mono-tonal attenuator can be used, of which an MPP is one. By the nature of our

material being a very thin porous mesh with micrometre sized holes, they can be viewed as approximately a micro-perforated plane, with only the mesh weave distinguishing it from a true MPP. (Lines 270-274)

14. The abbreviation of “rms” should be defined for clarity, and the variable D1 in equation (1) requires a clear explanation of its physical meaning.

We have added the requested information to the manuscript.(Lines 142-143, 334)

15. The sentence in line 257, “The absorption and attenuation coefficients of the material were unable to be measured at the,” is incomplete. Such errors should be thoroughly reviewed and corrected throughout the manuscript to ensure clarity and professionalism.

We apologize for making this mistake and have checked to make sure no mistakes of a similar nature are made in the manuscript.

16. The statement in line 258, “The high model accuracy should not necessarily be assumed without testing different measurement materials,” is vague. The authors should elaborate on why high accuracy should not be assumed and provide experimental evidence or reasoning to support it.

We agree this statement is vague, and perhaps we were not being strong enough with our claim and so have removed it, as MPP and thin-duct models are well studied within the literature and there is no particular reason they wouldn't be as accurate here. (Lines 361-372)

17. The thermographic measurement results presented in the manuscript are overly simplistic. To strengthen the findings, both quantitative results and perceptual validation should be included, providing a more comprehensive evaluation of the technique.

Perceptual validation was carried out through the SSIM metric, as well as in commentary around figure 2, which has been expanded to include non-single focal point data.

18. The manuscript lacks sufficient validation for the precision acoustic generation achieved by the proposed camera-in-the-loop holography algorithm, which is presented as a core contribution. Experimental demonstrations and quantitative analyses should be provided to substantiate its effectiveness.

We have added quantitative analysis through statistical tests to the existing data in order to substantiate it further. (Lines 741-742, 748-753)

19. The manuscript requires significant grammatical and structural improvements to enhance clarity, flow, and readability. A thorough review of the language and logical structure is recommended.

We have edited large swathes of the manuscript in this revision and we believe this has now been improved.

20. Several conclusions presented in the manuscript lack proper citations. To improve the scientific rigor and credibility of the work, the authors should add appropriate references to support their claims.

We have added citations where we noticed they were missing and hope any particular examples that may have been missed may be pointed out to us in further detail.

Reviewer 2:

In this work, the authors introduce a rapid 2D thermographic measurement technique, which is orders of magnitude faster than microphone scans, at the cost of acceptable losses in accuracy and phase information, with a maximum peak pressure of 12kPa measured, and a demonstrated average accuracy of 2.5% in peak measurement. The manuscript is well organized and the revision suggestions are as following.

It is suggested to address the problem raised in the first paragraph. One equation or one figure is expected for a better understanding of the principle of the holography in this work, to those who are not in this field. It is suggested to explain what are the hologram plane and the target plane, respectively. In optical holography, finding the exact solution of a desired hologram to reconstruct an accurate target object is an ill-posed inverse problem. Various non-convex optimization algorithms are thus designed to seek an optimal solution by introducing different constraints, frameworks, and initializations (Light: Science & Applications 13, 158 (2024)). Then it is suggested to explain why the Camera-In-The-Loop (CITL) Holography is used in this work. Thus, the motivation and the background of this work could be highlighted.

Thank you for this fantastic suggestion, we have added more introduction in the leading paragraphs to ground our problem statement in, including this excellent work you have suggested and some of the work cited within. (lines 49-65)

The real-time measurement and processing are important for the task. It is suggested to address the calculation time for the algorithms in this work. There should be a balance between the calculation speed and accuracy.

We have added the computation time to the work, noting it is minimal and in all cases lesser than the capture time of the data. (lines 626-641)

Reviewer 3:

General Comments:

The authors have presented a rapid 2D thermographic measurement technique that could be a valuable insight into the thermographic assessment of acoustic fields using gradient and steady-state methodologies. The author reported that the technique is faster than microphone scans but at the cost of acceptable losses in accuracy and phase information. Further, the technique is integrated with holography algorithms to establish a camera-in-the-loop algorithm that employs real-time measurement, enabling targeted data acquisition and on-line training of acoustic holography algorithms.

Overall, the manuscript is written and documented well.

Major Concerns:

There are several concerns listed below:

1. The major concern is the novelty of the proposed study compared to existing methods. The authors should discuss in detail the novelty and major findings.

We have clarified the novelty compared to existing methods highlighting that compared to the previous method in air, we model the measurement material, leading to a much more accurate calibration, which also allows optimisation of the measurement material for the maximum signal to noise ratio and sensitivity. In addition, we also attempt to model the impact acoustic streaming has on the steady-state case as well as conduction within the material, as well as looking at ways to increase the signal to noise ratio further, through various noise reduction methods. Finally, we also lay out that we present the first ever CITL system to use multiple pressure measurements at the same time, opening up the avenue of online holography algorithms for in-air acoustics. (Lines 869-886, table 4)

2. Does the proposed study consider all the following conditions:

- non-linear effects,
- ambient and device conditions,
- such as temperature[8, 9],
- humidity,
- device properties, such as the impulse response of transducers.

We consider all of these in varying ways:

- **Non-linear effects:** The absorption mechanism of sound into the mesh is inherently non-linear and so is considered by way of use on the MPP and thin-duct models. Streaming is also a non-linear effect and is again considered in this work. Harmonics are not however considered, but due to the narrow-band attenuation properties of the mesh, is not expected to affect the results too much.
- **Ambient conditions:** The temperature of ambient was taken from the thermal video files and was in the range to be expected from a HVAC controlled

university room of 19-22 degrees. Humidity was not measured but it is noted that both of these will have minor effects on our study at the levels of pressure we are measuring. For example, there is only a 3.6% difference in heat capacity of air between 0-100% humidity at 25 degrees Celsius and about a .3% change in speed of sound. (Lines 456-461) For temperature going from 20-30 degrees (wider than our measured conditions) does not change specific heat capacity to the third decimal place and only changes the speed of sound by about 2%. (Lines 496-504)

- Device conditions: The temperature of the phased array was monitored during measurement, with the temperature of the phased array being brought up to the same as that observed at the end of the microphone measurement when measuring with the thermal camera in order to control for it as best as possible and this was noted in the methods section of the manuscript.(Line 996-1003)
- Device properties such as impulse response are considered but are insignificant: The transducer impulse response is on the order of 1ms, which is drastically lower than the frame-time of the camera of 16/33ms @ 60/30 fps. We have added this information to the manuscript with citations for the impulse response time.(Lines 1004-1008)

3. The literature survey should be improved by citing the works on sound field imaging by digital holography, e.g., Prof. Awatsuji's group.

We agree and we have added a statement about Prof. Awatsuji's work to the introduction, noting how it has advanced on the optical methods listed, but that we are not directly competing against this method for the reasons listed before in the introduction of simplicity, cost and it's real-time nature due to the low processing and capture time, at the cost of the lack of phase information and the capability for high sensitivity for example. (Lines 111-113)

4. Detailed explanations of Fig. 1 and 2 should be provided in the text.

We have added these explanations in the manuscript.

5. The computation time to get the data in each case should be provided.

We have added the computation time to the work, noting it is minimal and in all cases lesser than the capture time of the data. (lines 626-641)

6. How does noise reduction play an important role in ensuring the gradient method? Quantitative analysis should be discussed.

We have added a more explicit explanation of the various parameters of the noise reduction and how they affect the accuracy and why, for example expanding upon the box filter mentioned in the comment below. (Lines 547-577)

7. The authors have used a 7×7 box filter average is used. Is it the optimized size?

Apologies for the mistake, an earlier version of the work used a 7x7 filter size, but a 5x5 filter was used in the end. We have added information comparing a 3x3, 5x5 and 7x7 filter to the paper, showing that for our data-set here the 5x5 was likely optimal. . (Lines 567-577)

Reviewer 4:

This paper proposes a fast two-dimensional ultrasonic holographic measurement technology based on thermal imaging. The measurement speed is significantly improved by the thermal gradient method and the steady-state temperature method, and the camera closed-loop (CITL) algorithm is combined to achieve high-precision real-time control of the sound field. The research has important application value in the fields of ultrasonic tactile feedback and acoustic levitation. The theoretical model is closely combined with experimental verification, and the overall innovation is strong. However, the paper has certain deficiencies in experimental details, data analysis, theoretical model verification, etc., which need to be further supplemented and improved. The following are specific review comments:

Combined with the CITL algorithm, real-time control of the sound field is achieved, providing a new technical path for applications such as ultrasonic tactile feedback and acoustic levitation. However, the paper's discussion of technical limitations is relatively superficial, and the impact of material melting threshold, nonlinear acoustic effects, etc. on high-pressure measurement is not deeply analyzed.

We have clarified that the limitations such as material melting are likely unable to be achieved in typical scenarios as they would likely happen above the saturation point of ultrasound in-air. We have expanded upon the non-linear limitations, focussing on the generation of higher harmonics which are not attenuated by our narrow-band absorber.(Lines 718-720)

The performance evaluation of the CITL algorithm is limited to single-point and multi-point focus, and it is recommended to expand it to more complex sound field distributions (such as dynamic sound fields or non-uniform sound fields).

Whilst the authors would love to apply this work to dynamic or non-focal point sound fields, we believe that would be outside the current scope of this work.

The experimental design is generally reasonable, but there is a lack of clear explanation of the number of experimental repetitions and sample size. It is recommended to supplement the statistical significance analysis of the experiment (such as t-test or ANOVA).

We have added information about the sample size of all tests, as well as adding Welch's t and Brown-Forsythe tests to the CITL experiments to validate the difference in mean and standard deviation. (Lines 530-535, 741-743, 748-753)

The paper does not discuss in depth the limitations of thermal imaging technology in high-pressure measurement (such as material melting and nonlinear acoustic effects). It is recommended to add relevant experimental data or theoretical analysis.

We have clarified that the limitations such as material melting are likely unable to be achieved in typical scenarios as they would likely happen above the saturation point of ultrasound in-air. We have expanded upon the non-linear limitations, focussing on the generation of higher harmonics which are not attenuated by our narrow-band absorber. (Lines 718-720)

The applicability of the CITL algorithm in complex sound fields is not discussed enough, and it is recommended to expand the experimental scenario to verify its robustness.

Whilst the authors would love to apply this work to dynamic or non-focal point sound fields, we believe that would be outside the current scope of this work. We have although added a thermal measurement of a non-focal point based sound field (Figure 2) in order to demonstrate that the measurement technique is robust to more complex sound fields, though we appreciate that this does not mean the CITL algorithm will as well.

Reviewer 5:

The manuscript presents a novel thermographic technique for rapid 2D measurement of in-air ultrasound holography, providing a significant improvement over conventional scanning microphone techniques. Additionally, the authors integrate this measurement approach with a camera-in-the-loop (CITL) holography algorithm to enhance real-time feedback and accuracy. The study is well-motivated, addressing a critical limitation in ultrasound-based haptics, volumetric displays, and acoustic holography. The results indicate orders-of-magnitude speed improvement

while maintaining reasonable accuracy, making this technique relevant for real-world applications. Although I acknowledge the idea and efforts behind the work I do have some concerns which I have shared in the following section.

1. Scientific Merit and Novelty

Strengths:

- The thermographic measurement method is novel and presents a compelling alternative to microphone-based approaches.
- The integration of real-time measurement with CITL holography improves algorithm adaptability and accuracy.
- The study accounts for critical factors such as non-linearity, acoustic streaming, and heat diffusion.

Weaknesses:

- While the method shows promise, a more comprehensive comparison with other established techniques such as Schlieren imaging, Laser Doppler Vibrometry (LDV), and heterodyne interferometry is needed to better establish its advantages and trade-offs.
- The validation of pressure measurements beyond 10 kPa remains uncertain due to the limitations of the reference microphone system.

Observation 1: The manuscript has some novelty with high scientific merit but requires stronger comparative analysis.

2. Technical Rigor and Methodology

Strengths:

- The experimental validation is thorough, with comparisons to conventional microphone-based approaches.
- The manuscript provides detailed modeling of thermographic pressure measurements, incorporating material properties and wave interactions.
- Quantitative performance metrics (RMSE, peak error, FWHM) are used effectively to support the method's validity.

Weaknesses:

- Some aspects of the mathematical modeling could be presented with more intuitive explanations for accessibility to a broader audience.
- The accuracy of measurements under extreme conditions (>10 kPa) is uncertain and requires additional validation.

Observation 2: The methodology is well-structured but would benefit from additional validation and clearer explanations of key modeling assumptions.

3. Impact and Suitability for Nature Communications Engineering

Strengths:

- The study introduces a promising breakthrough in ultrasound holography measurement, enabling fast and scalable data acquisition.
- Potential applications in mid-air haptics, volumetric displays, and non-contact

acoustic measurement make the study impactful.

- The real-time adaptability of CITL for algorithmic improvements aligns with modern engineering trends.

Weaknesses:

- The paper should discuss more explicitly how this technique can be adopted in industrial and commercial settings to be in alignment with the scope of the Nature Communications Engineering journal

Observation 3: The impact is significant within its niche but may not fully meet the broad interdisciplinary scope expected by Nature Communications Engineering.

4. Clarity and Presentation

Strengths:

- The manuscript is well-structured with a logical flow from problem statement to solution.
- Figures and tables effectively illustrate the methodology and results.
- The inclusion of a GUI for measurement validation enhances accessibility.

Weaknesses:

- Some sections, particularly the analytical modeling, are mathematically dense and could benefit from additional intuitive explanations.
- A summary table highlighting key findings would improve readability.
- Experimental details regarding calibration and data reproducibility should be explicitly stated.

Due to the open-source nature of the hardware and software used, the data should be reproducible. A thorough explanation of all techniques used is explained in addition to their implementation and calibration details have been further added to the methods. (Lines 967-1025, 1028-1040)

Observation 4: The manuscript is well-written but would benefit from improved clarity in technical explanations and structure.

Recommendation

After carefully evaluating the manuscript, I provide the following recommendation:

Major Revisions Required (Before Reconsideration for Communications Engineering)

Required Revisions:

1. Provide a stronger comparison with alternative measurement techniques (e.g., Schlieren imaging, LDV) to better position the method.

We have expanded upon our advantages stating that they are cost (a mesh can be bought for a few dollars and a cheap thermal camera can be had for around \$100-200), simplicity in terms of equipment pieces (no lenses, optical tables, beam splitters etc....), robust to alignment (vibrations for example can cause the beam to miss the knife edge in schlieren) as well as there being no need for computer tomography algorithms or phase-unwrapping. We do note our methods downsides in

comparison, for example that phase is not able to be measured, and the sensitivity and signal to noise ratio at low pressures is significantly worse. (Lines 169-183)

2. Clarify the accuracy and limitations of high-pressure measurements (>10 kPa) with additional validation.

We have clarified that validation of such measurements was not possible due to the limitations of the microphone, and that we do not have access to any other measurement technique which could help validation here. We have toned down our claims in response to this.

3. Improve accessibility of mathematical modeling by adding intuitive explanations where possible.

We agree and have added intuitive explanations to all those presented in the main introduction, results or discussion section (Lines 335-342, 444-448). We have not done the same for some of the methods section, as any explanation would increase the size of the manuscript drastically, and there are much better resources available to learn this information elsewhere.

4. Discuss the broader impact beyond ultrasound holography, emphasizing interdisciplinary relevance.

We have clarified the broader impact highlighting various commercial products beyond holpography itself including imaging (sonar), directional audio, haptics and acoustic levitation, all of which utilising ultrasound holography. (Lines 894-900)

5. Enhance clarity in presentation by restructuring dense sections and adding a summary table of key findings.

We have reworked much of the paper including the dense sections. In addition a summary table has been added to the discussion. (table 4)